# Evaluation of the New York State Mesonet Profiler Network Data

**Bhupal Shrestha[1], Jerald A. Brotzge[1] and Junhong Wang[1,2]**

[1]New York State Mesonet, SUNY University at Albany, Albany, New York

[2]Department of Atmospheric and Environmental Sciences, SUNY University at Albany, Albany, New York

***Correspondence to*****:** Bhupal Shrestha (bshrestha@albany.edu)

**Abstract.** The New York State Mesonet (NYSM) Profiler Network consists of 17 stations statewide. Each station operates a ground-based Doppler lidar (DL), a microwave radiometer (MWR) and an environmental Sky Imaging Radiometer (eSIR) that collectively provide profiles of wind speed and direction, aerosol, temperature, and humidity along with solar radiance, optical depth parameters and fish-eye sky images. This study presents a multi-year multi-station evaluation of Profiler Network data to determine the robustness and accuracies of the instruments deployed with respect to well-defined measurements. The wind speed (WS) measured by the DL and temperature (T) and water vapor density (WVD) measured by the MWR at three NYSM Profiler Network sites are compared to nearby National Weather Service radiosonde (RS) data while the aerosol optical depth (AOD) measured by the eSIR at two Profiler sites are compared to nearby in-situ measurements from the Aerosol Robotic Network (AERONET). The overall comparison results show agreement between the DL/MWR and RS data with a correlation of $R^2 \geq 0.89$ and between AERONET and eSIR AOD data with $R^2 \geq 0.78$. The WS biases are statistically insignificant and equal to 0 ($p > 0.05$) within 3 km whereas T and WVD biases are statistically significant and are below 5.5 ºC and 1.0 g m$^{-3}$, within 10 km. The AOD biases are also found to be statistically significant and are within 0.02. The performance of the DL, MWR and eSIR are consistent across sites with similar error statistics. When compared during three different weather conditions, the MWR is found to have slightly varying performance, with T errors higher during clear sky days while WVD errors higher during cloudy and precipitation days. To correct such observed biases, a linear regression method was developed and applied to the MWR data. In addition, wind shear from the DL and 14 common thermodynamic parameters derived from the MWR show an agreement with RS values with mostly $R^2 \geq 0.70$ and biases mostly statistically insignificant. A case study is presented to demonstrate the applicability of DL/MWR for

nowcasting a severe weather event. Overall, this study demonstrates the robustness, reliability, and value of the Profiler Network for real-time weather operations.

## 1 Introduction

The vertical profiles of winds, aerosols, temperature, and humidity are critical in understanding atmospheric exchange (physical and chemical) processes. Turbulence, friction, dispersion, vertical mixing, and transport lead to the exchange of heat, momentum and mass concentration ultimately affecting weather and air quality. Upper atmospheric data with high spatial and temporal resolutions are critical for operational meteorologists to assess and predict the atmospheric state. Various studies have shown the value of such data for improving nowcasting, short-range weather forecasting, and aviation services (Strauch et al., 1989; Wilczak et al., 1996; Shun et al., 2008; Chan et al., 2011; Madhulatha et al., 2013; Oude Nijhuis et al., 2018). Furthermore, the finer the temporal resolution of such data, the better the nowcasting of short-lived convective events (Feltz et al., 2002; Hu et al., 2019). As a result, forecasting centers are ingesting high resolution atmospheric profile data from the lower troposphere in real-time to provide more accurate forecasts of hazardous weather and air quality (Illingworth et al., 2019). However, there is a noted gap in observation within the boundary layer at high spatial and temporal resolutions (Wagner et al., 2019; Hu et al., 2019).

Recent advances in ground-based remote sensing profiling technology have spurred a plethora of new, large-scale deployments of lidars, microwave radiometers, sodars and ceilometers, such as the Sodar Network (Granberg et al., 2009), DWD Ceilometer Network (Thomas, 2017), Helsinki Testbed (Koskinen et al., 2011), E-PROFILE Network (Illingworth et al., 2019), and Unified Ceilometer Network (Delgado et al., 2020). These systems provide a ready means for monitoring atmospheric profiles at high temporal and spatial resolutions and under various weather conditions. Dense ground-based profiling networks have several advantages over the radiosonde network and satellite observations. Most global radiosonde stations launch radiosondes only twice daily (00 UTC and 12 UTC) and so fail to capture atmospheric variability through the entire the diurnal cycle (Wang and Zhang, 2008). Satellites provide global coverage filling gaps between stations where radiosonde measurements are unavailable, but the spatial and temporal resolutions of such measurements are low and are frequently impacted by the presence of clouds. Thus, large-scale ground-based networks of remote sensing profilers can complement

radiosonde and satellite systems, filling a critical need for lower tropospheric data sampling at high resolutions. But as these new profiler networks become increasingly common, it is important to assess the robustness, capability, and accuracy of these remote sensing instruments.

In order to test and evaluate the value of a network of vertical profiling systems for high-impact weather operations, the University at Albany, State University of New York, deployed the New York State Mesonet (NYSM) Profiler Network (Shrestha et al., 2021; www.nysmesonet.org/networks/profiler). The network consists of 17 ground-based stations deployed across the state between 2016 and 2018 (Fig. 1). Since then, the Profiler Network has been operating autonomously and continuously in real-time. Each station is comprised of a collocated scanning Windcube Doppler lidar (DL), a microwave radiometer (MWR) and an environmental sky imager and radiometer (eSIR) that collectively provides continuous real-time profiles of winds, aerosols, temperature, and humidity along with solar radiance, optical depth, and fisheye sky images. All data are collected, quality-controlled, and archived in real-time every 10 minutes. A detailed overview of the NYSM Profiler Network is presented in Shrestha et al. (2021). This paper focuses on evaluating the accuracy of the data collected from the NYSM Profiler Network with respect to well-defined reference measurements. The DL and MWR data are compared to National Weather Service (NWS) radiosonde data, while data collected from the eSIR are compared with in situ measurements from the Aerosol Robotic Network (AERONET).

Several studies have already assessed and evaluated the accuracies of data collected from DL and MWR that show correlation of $R^2 \geq 0.90$ and root mean squared error, RMSE $\leq 2.1$ m s$^{-1}$ for the DL wind speed measurements (Vermeesch et al., 2011; Kumer et al., 2014; Paschke et al., 2015; Dai et al., 2020; Mariani et al., 2020) and $R^2 \geq 0.98$ and RMSE $\leq 7$ K and $R^2 \geq 0.88$ and RMSE $\leq 2$ g m$^{-3}$ for the MWR temperature and water vapor density measurements respectively (Ware et al., 2003; Cimini et al., 2011; Madhulatha et al., 2013; Cimini et al., 2015; Xu et al., 2015; Bianco et al., 2017). The MWR's ability to measure relative humidity appears rather limited with $R^2 \geq 0.48$ and RMSE $\leq 25\%$ (Bianco et al., 2017; Xu et al., 2015) as the MWR fails to capture the high-resolution vertical details of the water vapor due to its coarser resolution.

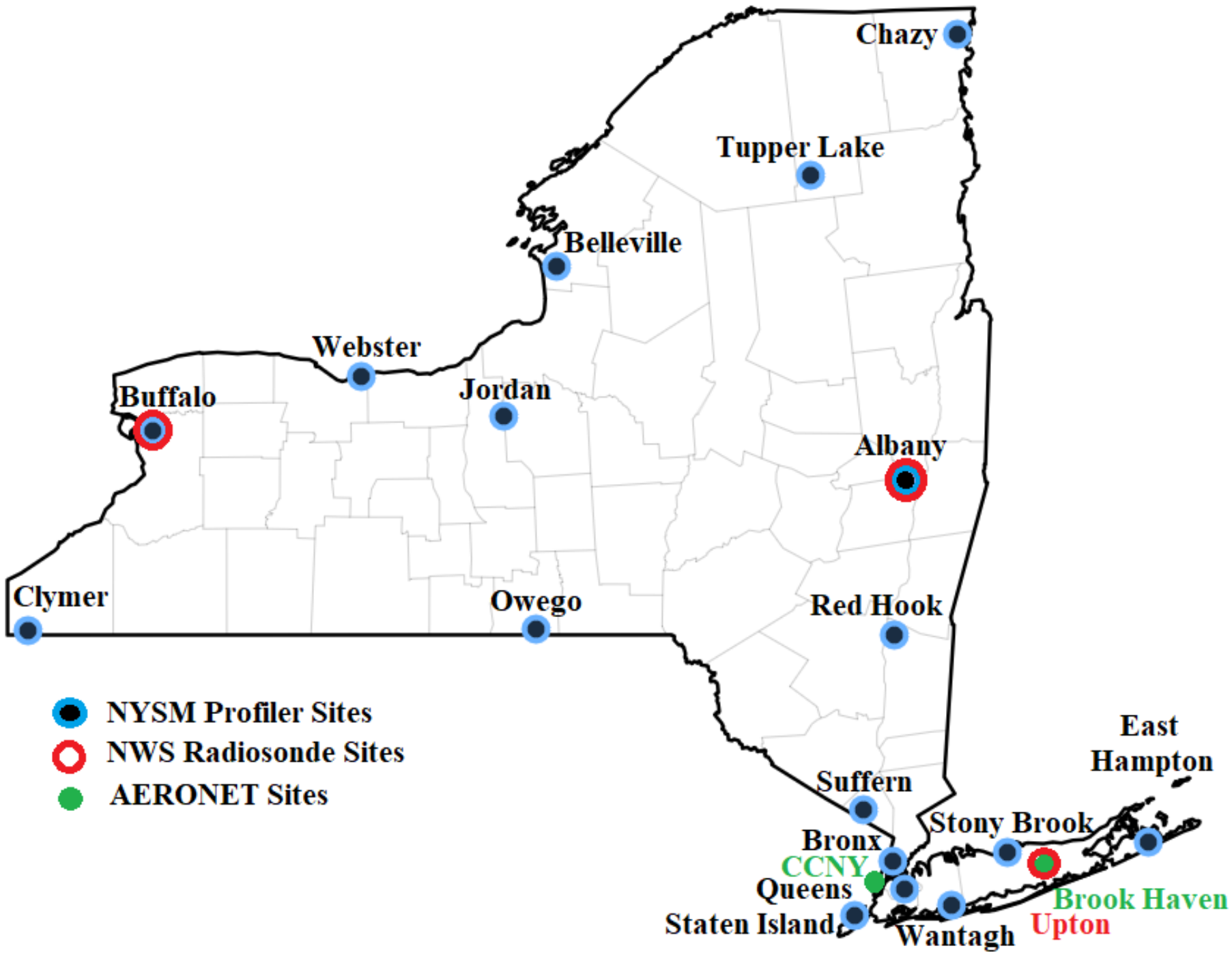

Figure 1. A map of New York State Mesonet Profiler Network along with the NWS Radiosonde and AERONET sites.

Though most studies have shown high value for $R^2$ for most variables, a closer inspection of these prior results show marked variations in errors. Furthermore, many prior studies present results from just a few case studies limited to a few days to few months or seasons, generally not exceeding a year and usually from a single site. Thus, those results could be influenced by local topography, seasonal variations, and other local factors and some potentially due to the varying operational procedures and retrieval methods used. The aim of this study is to build on the results from previous studies but by using a much broader and more extensive dataset. This review evaluates the accuracy of data collected from three different NYSM profiler sites that are located near NWS radiosonde sites, namely Buffalo (urban), Albany (Upper Hudson Valley) and Stony Brook (coastal) representing upstate, central, and downstate regions of the state respectively, during the period from January 2018 to August 2021. This multi-station multi-year study provides a comprehensive evaluation of the performance and robustness of the instruments from across different topographical regions and meteorological conditions. Next, this study presents an

evaluation of derived parameters such as wind shear from the DL and convective (thermodynamic) parameters from the MWR. The accuracy of these derived parameters demonstrates the suitability of the DL and MWR for use in real-time weather applications, which is severely limited with traditional twice daily radiosonde data. Lastly, this study evaluates the aerosol optical depth (AOD, a widely used parameter in air quality studies and forecasting) as derived from the eSIR at two NYSM profiler sites located at Stony Brook and Bronx where AERONET sites are located nearby. Overall, this paper provides quantification and understanding of observational errors associated with profiler network data based on well characterized in-situ measurements from the NWS radiosondes and AERONET (see Fig. 1) that are critical for several weather and air quality studies and forecasting.

This paper is structured as follows. Section 2 provides a summary of the instrumentation and siting of the NYSM Profiler Network, NWS Radiosondes and AERONET. Section 3 reviews the data and methodology, followed by results and discussions of the evaluation of the data in Section 4. A summary and conclusions are presented in Section 5.

## 2 Instrumentation and experimental sites

Each of the 17 NYSM Profiler Network stations is comprised of an active remote sensing Leosphere-Vaisala Scanning Windcube Doppler lidar (DL) 100S, a passive remote sensing Radiometrics MP-3000 series Microwave Radiometer (MWR) and an in-house built Environmental Sky Imaging Radiometer (eSIR, commonly referred to as a sun photometer) (Shrestha et al., 2021). These three instruments are collocated at each of the 17 profiler sites. Most profiler sites (except Albany, East Hampton, and Webster) are located within 0.5 km of a NYSM Standard Network site that provides atmospheric data at or near the surface (Brotzge et al., 2020).

The DL operates at the near-infrared 1540 nm wavelength and provides radial wind speed and Carrier-to-Noise Ratio (CNR, a modulated signal for Signal-to-Noise Ratio, SNR) using a highly sensitive heterodyne detection technique (Boquet et al., 2016). The DL is operated in Doppler Beam Swinging (DBS) mode (Newman et al., 2016); the DBS points in five directions (four cardinal direction scans at an elevation of 75º and one vertical 90º scan) which are averaged together to yield the 3D (u, v, and w) wind speeds. The measurement is from 100 m to 7000 m with vertical resolution of 25 m below 1000 m and 50 m above 1000 m and with a temporal resolution of ~ 20s (1 DBS scan).

The MWR operates in the 21 K band (22-30 GHz) and 14 V band (51-59 GHz) channels to measure brightness temperatures in the water vapor and oxygen bands that are then converted into profiles of temperature, relative humidity, water vapor density and liquid density using a neural network and radiative transfer algorithm (Solheim et al., 1998; Ware et al., 2003; Knupp et al., 2009). The measurement is from the surface to 10 km with vertical resolution of 50 m below 500 m, 100 m between 500 m and 2000 m and 250 m above 2000 m, and with temporal resolution of approximately 2 min.

The eSIR operates a shadow band technique (Harrison et al., 1994) and measures spectral direct and diffuse irradiance at seven wavelength channels (415, 500, 610, 670, 870, 940 and 1020 nm) every 5 minutes during daylight hours. Additionally, it also provides fish-eye sky images and has a GPS, temperature, pressure, and humidity sensors. Measurement accuracies provided by the sensor manufacturers and reference measurements are listed in Table 1.

Table 1. Measurement accuracies reported by the manufacturer

| | **Wind Speed (m s$^{-1}$)** | **Wind Direction (°)** |
|---|---|---|
| Doppler Lidar | 0.5 | 2.0 |
| NWS Radiosonde LMS-6 | 1.0 | 5.0 |
| NWS Radiosonde Vaisala RS92 | 0.15 | 2.0 |
| | **Temperature (°C)** | **Relative Humidity (%)** |
| Microwave Radiometer | 0.5 | 2.0 |
| NWS Radiosonde LMS-6 | 0.3 | 5.0 |
| NWS Radiosonde Vaisala RS92 | 0.5 | 5.0 |
| | **AOD** | |
| Sun Photometer | ~ 2 – 4 % | |
| AERONET (Level 2.0) | ~ 2 % | |

Three NWS Radiosonde (RS) sites operate across New York State – Buffalo (BUF), Albany (ALB) and Upton (OKX). The radiosondes are launched twice daily at 00 UTC and 12 UTC (7 p.m. and 7 a.m. EST) and provide vertical profiles of pressure, temperature, relative humidity, dew point temperature, wind speed and direction from the surface to around 30 km AGL at about 1s temporal resolution. The NWS launches the Lockheed Martin LMS-6 radiosonde at Buffalo and Albany, and Vaisala RS-92 at Upton (measurement accuracies listed in Table 1). These three radiosondes launch sites – Buffalo, Albany, and Upton, are located near the NYSM Profiler Network sites and hence, the radiosonde data are compared against the DL and MWR data at Buffalo, Albany, and Stony Brook, respectively. The AERONET has a few sites in New York, with all sites located around the New York city region. Two sites – Brookhaven and CCNY are in

close proximity to the NYSM Profiler Network sites at Stony Brook and Bronx, respectively. The pre- and post-calibrated, cloud screened and quality assured level 2.0 data from the AERONET are used for comparison with the eSIR data. The details about AERONET level 2.0 AOD and data processing can be found in Giles et al. (2019). The New York state map with the NYSM Profiler Network, NWS radiosonde and AERONET sites are shown in Fig. 1 and the location of the selected sites, and their average separation distances are listed in Table 2.

Table 2. NYSM, NWS and AERONET site information

| NYSM Site | Location (Lat, Lon) | NWS Site | Location (Lat, Lon) | Separation Distance (km) |
|---|---|---|---|---|
| Buffalo | 42.99, -78.79 | Buffalo (BUF) | 42.94, -78.72 | 8 |
| Albany | 42.75, -73.81 | Albany (ALB) | 42.69, -73.83 | 7 |
| Stony Brook | 40.92, -73.13 | Upton (OKX) | 40.87, -72.86 | 24 |
| **NYSM Site** | **Location (Lat, Lon)** | **AERONET Site** | **Location (Lat, Lon)** | **Separation Distance (km)** |
| Stony Brook | 40.92, -73.13 | Brookhaven | 40.87, -72.88 | 22 |
| Bronx | 40.87, -73.89 | CCNY | 40.82, -73.95 | 7 |

## 3 Data and methodology

The high-resolution NWS radiosonde data are downloaded from the University of Wyoming archive (http://weather.uwyo.edu/upperair/bufrraob.shtml) and have a vertical resolution of 1s, equivalent to ~5m. The radiosonde profiles of temperature, water vapor density and wind speed from January 2018 to August 2021 are considered in this study. A total of 2093, 2457 and 1862 NWS radiosonde profiles were available during the times when the MWRs at Buffalo, Albany and Stony Brook were operating but based on the MWR data availability (QA flag), a total of 2010, 2360 and 1755 pair of profiles have been selected for comparisons. On average ~96% of profiles were available from the MWR for comparison with the radiosondes. Similarly, a total of 2165, 2655, 2408 radiosonde profiles were available during the times when the DL at Buffalo, Albany and Stony Brook were operating but based on the Doppler lidar data availability, a total of 1752, 1953 and 2109 pair of profiles have been selected for comparisons. Since the aerosol concentration, atmospheric refractive turbulence, humidity, and precipitation have significant impact on the data availability from the Doppler lidar (Aitken et al., 2012), the total number of profiles selected for RS-DL comparisons are relatively lower than that of RS-MWR comparisons. Furthermore, since the DL data availability is determined by the CNR threshold (Boquet et al., 2016) and CNR values are dependent on the aerosol concentration, the CNR values typically

follow the diurnal cycle with lower values at night reaching local minimum in early morning and higher values during the day (Aitken et al., 2012). This results in lower data availability at night and morning and higher data availability during the day and evening. Thus, the DL data availability particularly during the morning NWS radiosonde launch time (7 a.m. LT or 12 UTC) is not optimal and is usually lower than at other times (Fig. 2), thereby reducing the number of RS-DL profiles for comparisons. Nevertheless, on average ~80% profiles were available from the DL for comparison with the radiosondes. The major data gaps for each instrument during the comparison period are listed in Table 3.

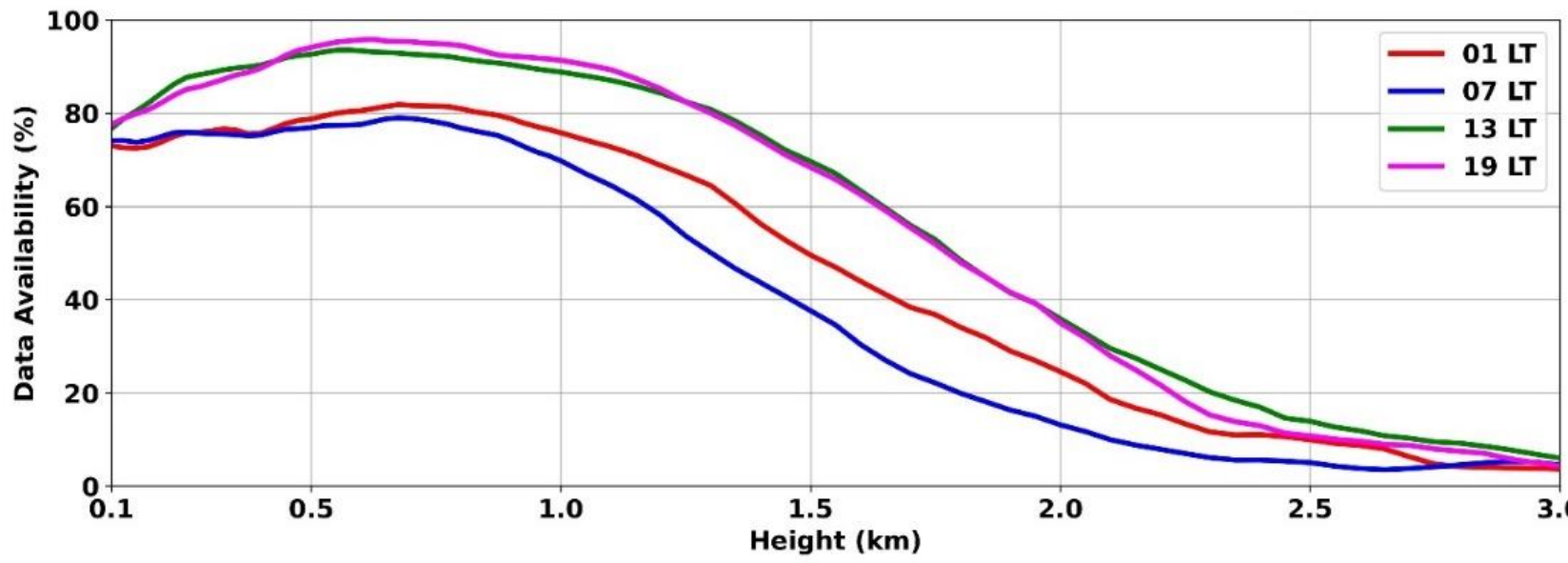

Figure 2. The Doppler lidar data availability during four different hours at NYSM Profiler site at Albany during May – August 2021. The NWS radiosonde launches are at 12 UTC (7 LT) and at 00 UTC (19 LT).

Table 3. Major data gaps from January 2018 to August 2021

| Site | Sensor | Period | Reason |
|---|---|---|---|
| Buffalo | Microwave Radiometer | 09/17/2019 – 05/31/2020 | Roof repairs at host location |
| | Doppler Lidar | 09/17/2019 – 05/31/2020 | Roof repairs at host location |
| Albany | Microwave Radiometer | 01/28/2018 – 05/09/2018 | Failed k-band noise diode |
| Stony Brook | Microwave Radiometer | 10/11/2018 – 07/26/2019 | Failed v-band noise diode |

Since the radiosonde measurements have a finer vertical resolution (~5m) than that from the DL and MWR measurements; it is necessary to define a common height grid to make data comparable. To do so, radiosonde data within ±5 m of the DL/MWR measurement height are first averaged to smooth the radiosonde data. Since the data availability from the DL decreases with height (Fig. 2) due to its dependence on aerosol concentration, comparison data are usually limited

to within the boundary layer (BL) as the BL typically has more aerosols than the free troposphere. Therefore, the RS-DL data are only compared from 100 m to 3 km AGL. A typical the radiosonde has an ascent rate of 5m $s^{-1}$, which takes approximately 10 minutes to reach the height of 3 km. So, the horizontal wind speed profiles from the DL are averaged ±10 min centered at the radiosonde launch time and then compared with the corresponding profiles from the radiosonde. Similarly, the temperature and water vapor density profiles up to 10 km from the MWR are first averaged ±30 min centered at the radiosonde launch time and then compared with the corresponding profiles from the radiosonde. Since off-zenith (20º elevation) observations from the MWR provide more accurate measurements than zenith observations (Xu et al., 2014), the average of two off-zenith observations are used for the comparisons.

The MWR measurements have errors associated with its neural network retrieval technique (Cimini et al., 2011). The neural network is trained with RS data from a site with a similar altitude and climatology to the MWR site (Knupp et al., 2009). However, only three radiosonde sites are available in NYS and are far from most of the MWR site locations. This distance between the MWR and RS site limit the effectiveness of the neural network method and introduces some inherent error due to local climatology (Cimini et al., 2011). Additional error between the RS and MWR arises due to the large RS drift distance. Xu et al. (2015) has reported that MWR biases are height dependent, associated with wind speeds and the RS drift distances. Since the MWR is a ground-based instrument measuring along the line of sight while the radiosonde measures along the trajectory as it ascends and drifts horizontally with the winds, the two instruments may not sample the same air masses spatially when the RS drift distances are very large. Based on the data from Albany during the period of study, the RS was found to drift significantly with height. As wind speeds increase with height, the drift distances increase from 1.6 km at a height of 1 km to 6 km at a height of 3 km and 42 km at a height of 10 km. In addition to this spatial mismatch, there is also a temporal mismatch as the RS typically takes about 30 min to reach a 10 km height; however, the temporal mismatch is somewhat compensated by averaging the MWR data centered at the RS launch time. The spatial and temporal mismatch have less impact on the DL data because a maximum height of only 3 km is considered. In summary, horizontal wind speed profiles up to 3 km from the DL and temperature and water vapor density profiles up to 10 km from the MWR are compared against profiles measured by the RS. Data are further evaluated under three different weather conditions: precipitation, cloudy and clear sky days.

In addition to the directly measured data comparisons, several derived forecasting parameters from the DL and MWR are calculated and compared against those derived from the RS. Wind shear (100 m – 1 km and 100 m – 3 km) are derived from the DL using the horizontal wind speeds at the two height levels. Fourteen different thermodynamic parameters are derived using the MWR data. To calculate and compare the thermodynamic parameters, RS and MWR data are subsampled to a common pressure grid at 10 hPa resolution. A cubic spline interpolation is applied at 10 hPa intervals from the surface to the lowest pressure level available. Interpolation is specifically needed to make sure data are available at mandatory pressure levels as defined by the American Meteorological Society (2014). The thermodynamic parameters considered in this study are as follows:

(a) Moisture parameters – mean relative humidity (meanRH) and total precipitable water (TPW). The meanRH is calculated from the near-surface pressure level (**ps**) to 950, 850 and 700 hPa. The TPW (total water content present in the vertical column of air) is calculated as defined by Solot (1939) from **ps** to the lowest pressure ($\mathbf{p_L}$) available. The lowest pressure is normally equal to or lower than 300 hPa.

(b) Potential temperature ($\boldsymbol{\theta}$) lapse rate (LR) between **ps** and 850 hPa and **ps** and 700 hPa.

(c) Stability index – difference between the saturated equivalent potential temperature ($\boldsymbol{\theta_{es}}$) and equivalent potential temperature ($\boldsymbol{\theta_e}$) at two levels – 950 and 850 hPa.

(d) Thickness layer between **ps** and 850 hPa and **ps** and 500 hPa.

(e) Single-level indices such as K Index (KI), Lifted Index (LI), Showalter Index (SI) and Total Totals Index (TT). Details about these indices, their formulas and threshold values for severe convective weather forecasting can be found at https://www.weather.gov/lmk/indices.

Finally, the AERONET level 2.0 AOD data are downloaded from https://aeronet.gsfc.nasa.gov/new_web/aerosols.html. The AOD are derived from the eSIR using the Beer-Lambert-Bouguer Law and Langley regression (Koontz et al., 2013). Since the eSIR data are available every 5 minutes, eSIR-derived AOD are compared against 5-minute averaged AERONET AOD for the three commonly available wavelengths: 500 nm, 870 nm, and 1020 nm. Since there were only limited time periods when both eSIR and Aeronet data were available, the AOD data are compared from April to June of 2018 at Stony Brook and from March 2018 to October 2019 at Bronx.

The comparison statistics calculated between the reference measurements (radiosondes and AERONET) and NYSM Profiler Network measurements include slope (m), coefficient of determination ($R^2$) and three types of errors: mean bias error (MBE), mean absolute error (MAE) and root mean square error (RMSE).

## 4 Results and discussions

### 4.1 Evaluation of DL data

A comparison of horizontal wind speeds (WS) from RS and DL for the three sites (Buffalo, Albany, and Stony Brook) show high values for m and $R^2$, i.e., m ≥ 0.93 and $R^2 \geq 0.89$ (Fig. 3 a – c), implying good agreement between the two instruments. The DL shows very small to no biases across the sites and are within the expected range based on the accuracies of the DL and RS listed in Table 1. Such observed biases are in statistical agreement (statistically equal to 0, based on t-test, $p > 0.05$) at Buffalo while the biases are statistically significant and different from 0 ($p \leq 0.05$) at Albany and Stony Brook. The MAE ranges between 1.0 and 1.4 m $s^{-1}$ while the RMSE ranges between 1.4 and 1.9 m $s^{-1}$ across the three sites. Errors are found to be relatively higher at Stony Brook than at the other two sites. Across all three sites, differences are within 0.5 m $s^{-1}$, showing a consistent performance from the DL.

Error statistics and $R^2$ are plotted as a function of height (Fig. 3 d – f). Along the height, the MBE is very close to 0 and are in statistical agreement ($p > 0.05$) except for the lowest three heights (100, 125, 150 m) at Albany. Both at Buffalo and Albany, the MAE and RMSE are below 1.4 and 2 m $s^{-1}$ throughout the profile while at Stony Brook, the MAE and RMSE are below 1.8 and 2.4 m $s^{-1}$. The RMSEs are ≥ 2 m $s^{-1}$ mostly above 2 km at Stony Brook. Overall, WS errors (MBE, MAE and RMSE) are slightly larger at Stony Brook than at Buffalo and Albany which is consistent with those observed in Fig. 3 (a – c). Such relatively higher errors at Stony Brook could be due to the greater distance between the RS and DL locations, topographical differences, and the potential influence of the marine boundary layer. The NYSM site at Stony Brook is close to the coastal area (~2km) of the Long Island Sound while the corresponding NWS site at Upton is situated more inland and midway between the northern and southern coasts (~10 km, see Fig. 1 for approximate location). The $R^2$ profiles show the lowest value at 100 m that rapidly increase up to 0.5 km. Above 0.5 km, the $R^2 > 0.90$ at Buffalo and Albany and near or above 0.90 at Stony Brook. The overall lower values of $R^2$ within 0.5 km is consistent with studies by Mariani et al. (2020) and Kumer et

al. (2014). These lower correlations could be due to large uncertainties in radiosonde wind measurements below 0.5 km as a result of larger self-induced irregular balloon motions in the turbulent layer (Wang et al., 2009). Overall, the DL is able to capture the vertical structure of WS consistent with RS measurements as shown in the representative example in Fig. 4.

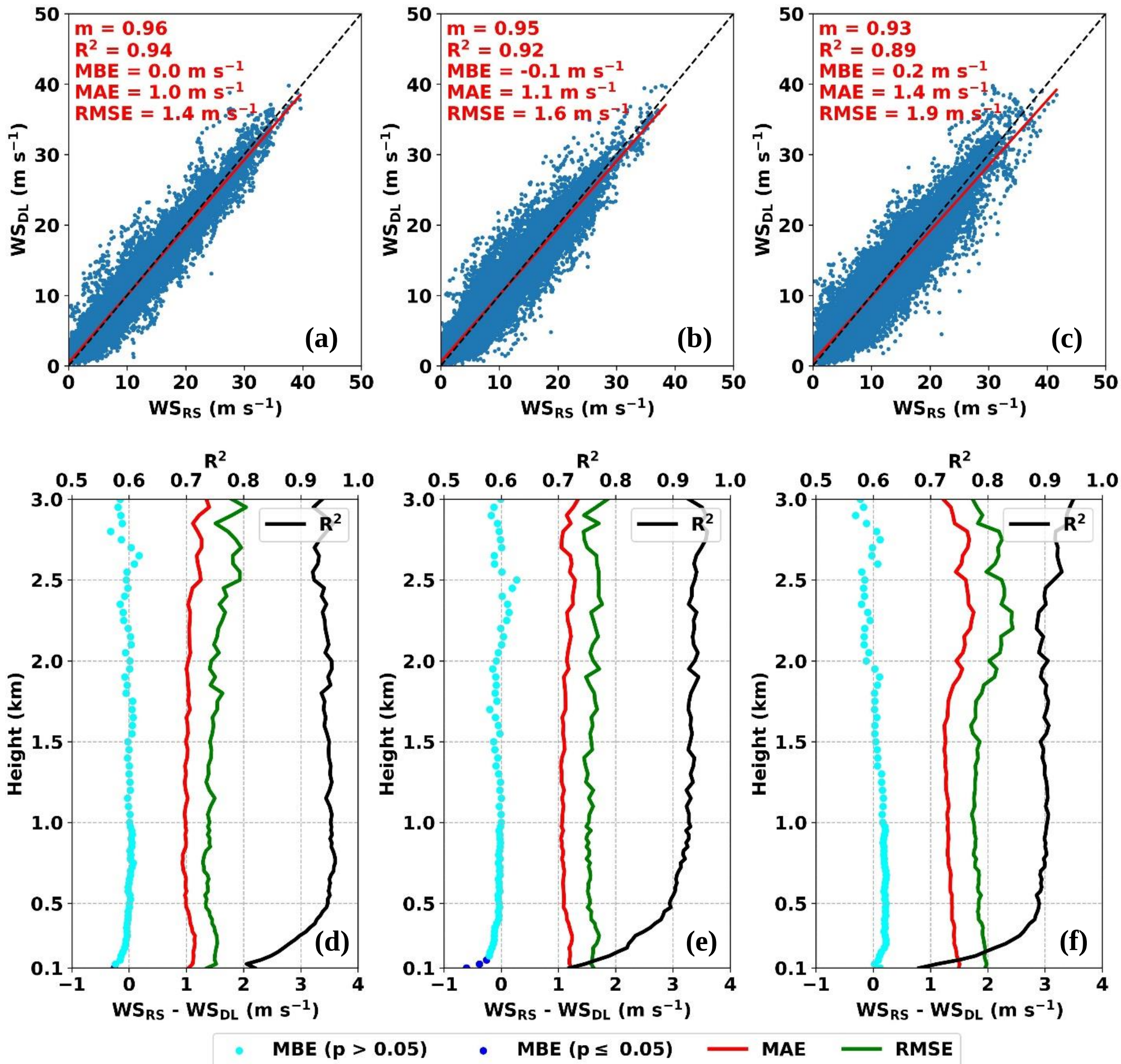

Figure 3. Scatterplots for RS and DL measured horizontal wind speed (WS) at three NYSM Profiler Network sites at: (a) Buffalo, (b) Albany, and (c) Stony Brook. Vertical profiles of $R^2$, MBE, MAE and RMSE for the same variable at the respective sites: (d) Buffalo, (e) Albany, and (f) Stony Brook.

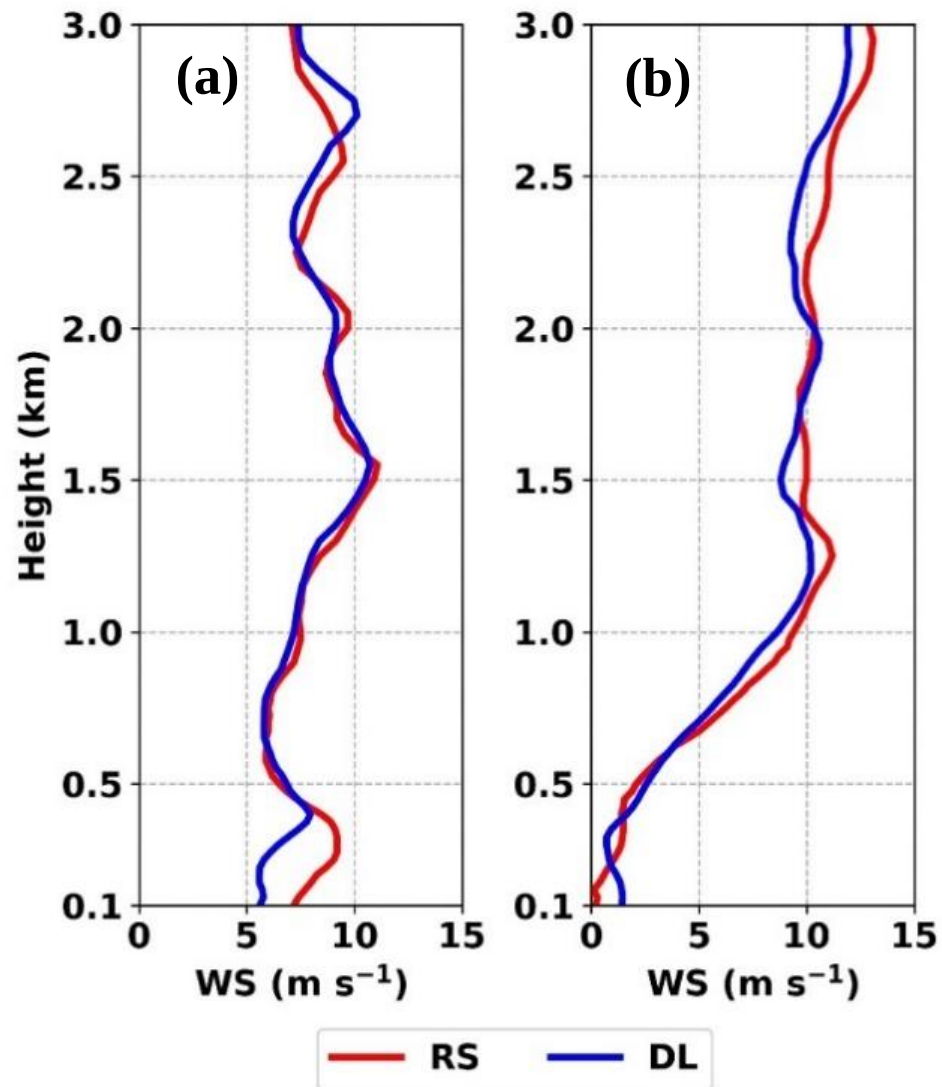

Figure 4. Vertical profiles of horizontal wind speed (WS) measured by DL and RS at NYSM Profiler sites at (a) Buffalo and (b) Albany at 19 LT (23 UTC) on 19 July 2021.

### 4.2 Evaluation of DL derived wind shear

Scatterplot comparisons of the RS and DL derived wind shear as calculated from 100 m to 1 km and 100 m to 3 km are shown for all three selected sites (Fig. 5 a – f). A total of 712 (41%), 848 (43%) and 951 (45%) profiles were available at Buffalo, Albany, and Stony Brook for the calculation of 100 m – 1 km wind shear. The $R^2 \geq 0.86$ are observed at Buffalo and Albany but only $R^2 = 0.70$ at Stony Brook (Fig. 5 a – c). The MBE of 0.5 m $s^{-1}$ at Albany is statistically significant and different from 0 ($p \leq 0.05$) while the MBE ≤ 0.2 m $s^{-1}$ at Buffalo and Stony Brook are in statistical agreement and difference equal to 0 ($p > 0.05$). The MAE ranges between 1.4 and 1.9 m $s^{-1}$ and the RMSE ranges between 1.7 and 2.4 m $s^{-1}$. The slightly larger MAE and RMSE at Stony Brook could be due to the influence of the nearby marine surface layer. Difference errors among sites are within 0.7 m $s^{-1}$.

A total of 94 (5%), 54 (3%), and 57 (3%) profiles were available at Buffalo, Albany, and Stony Brook for the calculation of 100 m – 3 km wind shear. Limited aerosols and attenuation limited the frequency of data availability from the DL at 3 km. The comparison results for the 100 m – 3 km wind shear show $R^2 \geq 0.88$ across all three sites (Fig. 5 d – f) with an increase in $R^2$ by 4.6 % (Buffalo), 2.3 % (Albany) and 31.4 % (Stony Brook) when compared to the 100 m – 1 km

wind shear (Fig. 5 a – c). Across three sites, the MBEs are found to be statistically equal to 0 (p > 0.05) and the MAE ranges between 1.7 and 1.9 m s$^{-1}$ while the RMSE ranges between 2.1 and 2.2 m s$^{-1}$, with differences among sites limited to within 0.2 m s$^{-1}$.

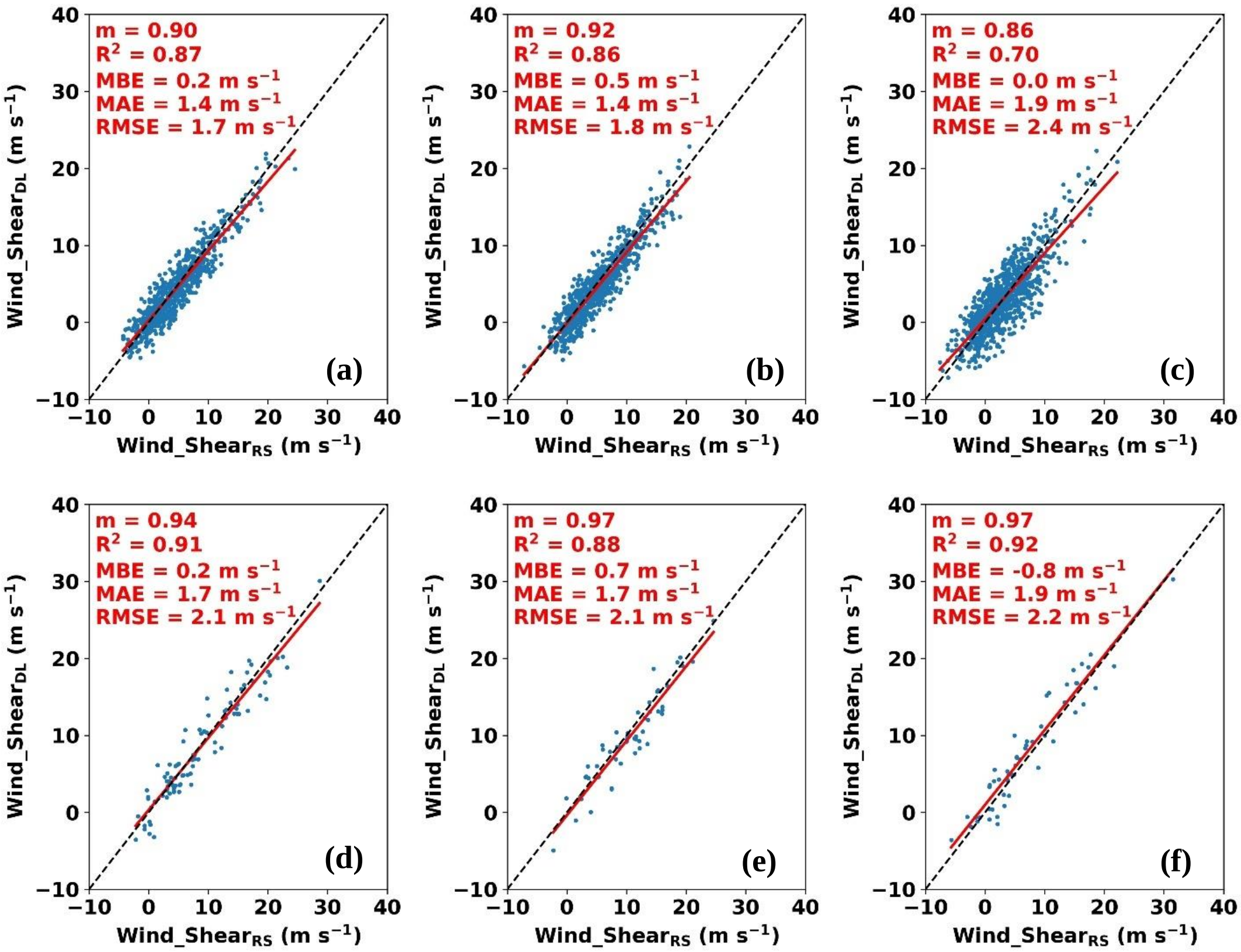

Figure 5. Scatterplots for RS and DL derived 100 m – 1 km wind shear at three NYSM Profiler Network sites at: (a) Buffalo, (b) Albany, and (c) Stony Brook and 100 m – 3 km wind shear at the respective sites: (d) Buffalo, (e) Albany and (f) Stony Brook.

For operational use, it is important to note that because of the DL dependency on aerosol concentration and meteorological conditions, the availability of DL data decreases with height, and therefore, the wind measurements at 3km AGL may be relatively difficult to obtain. As shown in Fig. 2, the DL data availability at 1 km is above 70% while at 3 km, is below 10%.

### 4.3 Evaluation of MWR data

A comparison of temperature (T) from the RS and MWR for three sites shows m ~ 1 and $R^2 \geq 0.97$ (Fig. 6 a – c). Across the three sites, the MWR shows significant cold biases (positive MBE), with the MBE ranging between 2.7 and 3.3 ºC. These cold biases are statistically significant ($p \leq 0.05$). The MAE ranges between 3.0 and 3.7 ºC and the RMSE ranges between 3.8 and 4.8 ºC. Site-to-site error differences are within 1 ºC, showing consistent behavior by the MWRs in measuring temperature. Error statistics are presented in Fig. 6 (d – f) as a function of height and show very similar vertical structures from one site to another. All three sites show $R^2 > 0.90$ within the lowest 2.5 km and $R^2 > 0.80$ below 7.5 km. The MWR shows cold biases in temperature throughout the profile and are statistically different from 0 ($p \leq 0.05$) except for a few lower heights. The observed cold biases are consistent with previous studies by Cimini et al. (2011), Xu et al. (2015) and Cimini et al. (2015). The MBE, MAE and RMSE increase rapidly within the boundary layer and reach as high as 5.4 ºC at 2 km. Above 2 km, both MAE and RMSE only vary within ~1 ºC. RMSEs are >4 ºC above 2 km at all three sites and sometimes exceed 6 ºC as seen at Albany. In general, the MWR follows the overall vertical temperature structure as measured by the RS; however, the MWR consistently fails to detect the elevated temperature inversion layer (Fig. 7 a – c). This causes a marked increase in cold biases above the layer. Such cold biases will have significant adverse impacts on operational applications, such as determining precipitation type and forecasting indices that rely upon temperature. Therefore, a simple correction method is developed and discussed in Section 4.5 to minimize such cold biases in MWR temperature.

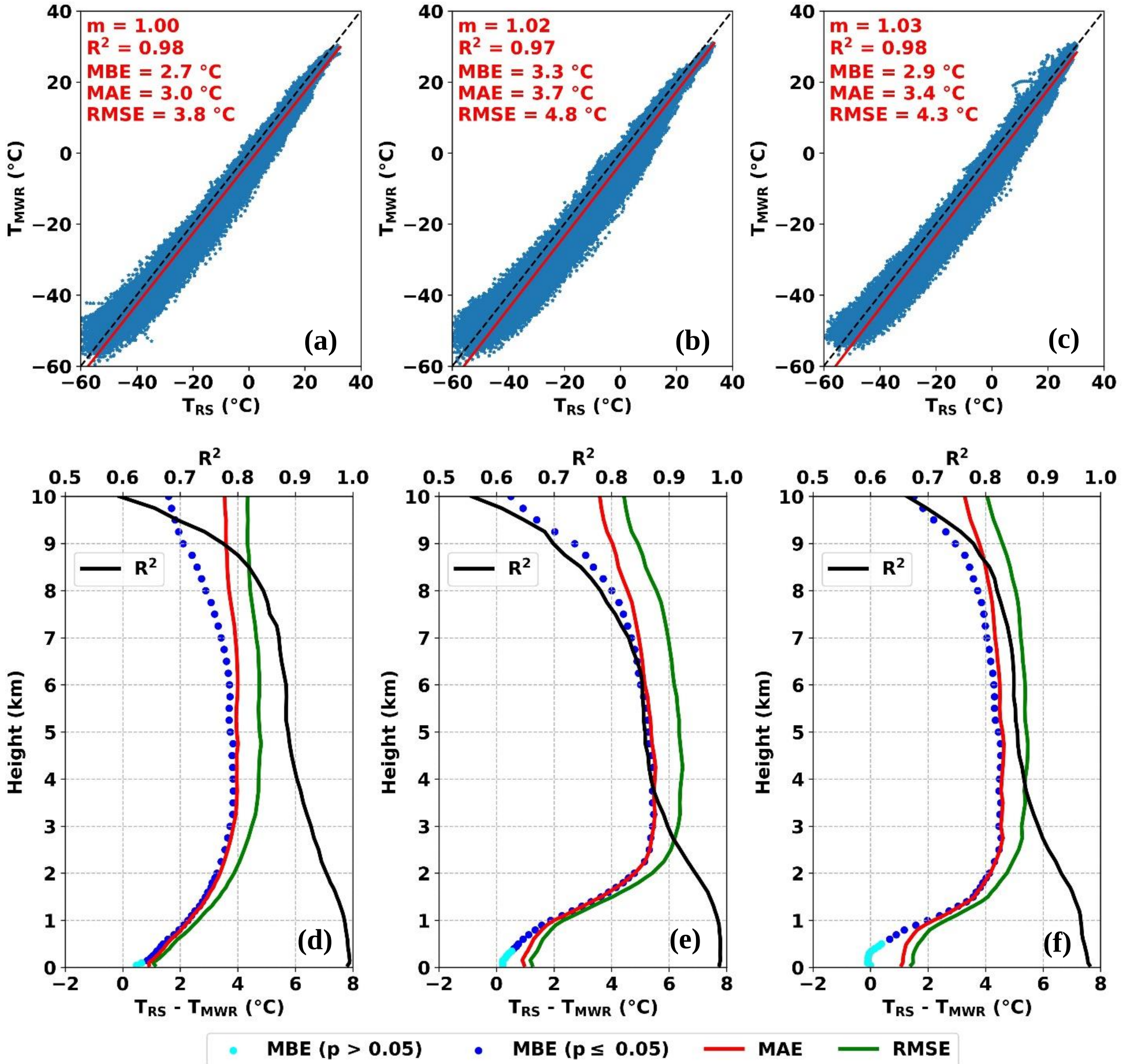

Figure 6. Scatterplots for RS and MWR measured temperature (T) at three NYSM Profiler Network sites at: (a) Buffalo, (b) Albany, and (c) Stony Brook. Vertical profiles of $R^2$, MBE, MAE and RMSE for the same variable at the respective sites: (d) Buffalo, (e) Albany, and (f) Stony Brook.

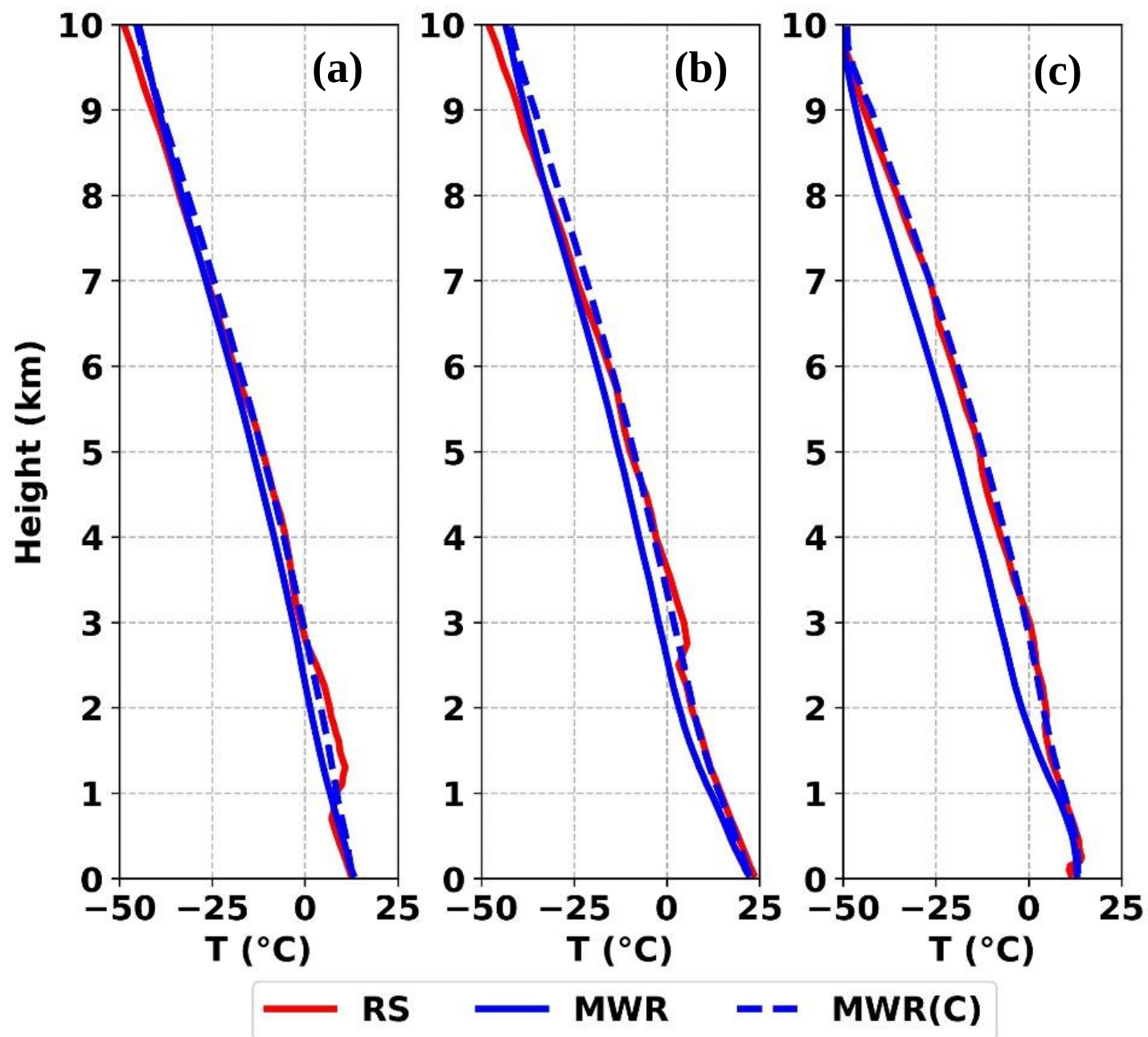

Figure 7. Vertical profiles of temperature (T) measured by RS and MWR – Original and Corrected (C) at (a) 23 UTC on 2 May (precipitation day), (b) 23 UTC on 28 July (cloudy day), (c) 11 UTC on 2 May 2021 (clear sky day) at Albany.

A comparison of water vapor density (WVD) from the RS and MWR are presented in Fig. 8 (a – c), for the three sites. Results show values of m ≥ 0.88 and $R^2 \geq 0.95$. The MWR results indicate dry biases (positive MBE) at Buffalo and Stony Brook that are statistically significant ($p \leq 0.05$) but the low wet bias at Albany is statistically insignificant ($p > 0.05$). The MAE ranges between 0.51 to 0.77 g $m^{-3}$, and the RMSE ranges between 0.79 and 1.19 g $m^{-3}$, both being higher at Stony Brook. Site error differences vary within 0.40 g $m^{-3}$, showing spatial consistency in the MWR measurements. The WVD error and $R^2$ as a function of height is presented in Fig. 8 (d – f). The MBEs are mostly statistically significant along the height. The MWR shows a dry bias below ~2 km that changes to a wet biases above ~2 km, with little bias observed above ~6.5 km. Such characteristic changes from dry to wet biases are consistent with Xu et al. (2015). Typically, errors are found to be largest within ~2 km where the MWR indicates a dry bias. The vertical profiles of MAE and RMSE show values greater than 0.5 g $m^{-3}$ below 4 – 6 km and below 0.5 g $m^{-3}$ above that height, similar to Cimini et al. (2015) and Xu et al. (2015). Within the boundary layer (below

~2km), errors are relatively larger at Stony Brook than the other two sites, which could be due to the influence of the moisture from the local marine boundary layer. The $R^2$ decreases with height with $R^2 \geq 0.90$ below 1 km and $R^2 \geq 0.80$ below 3 km across all three sites. The MWR tends to follow the general trend of the vertical structure of the WVD as measured by the RS; however, it consistently fails to capture the high-resolution vertical details, primarily due to its coarser resolution (Fig. 9 a – c).

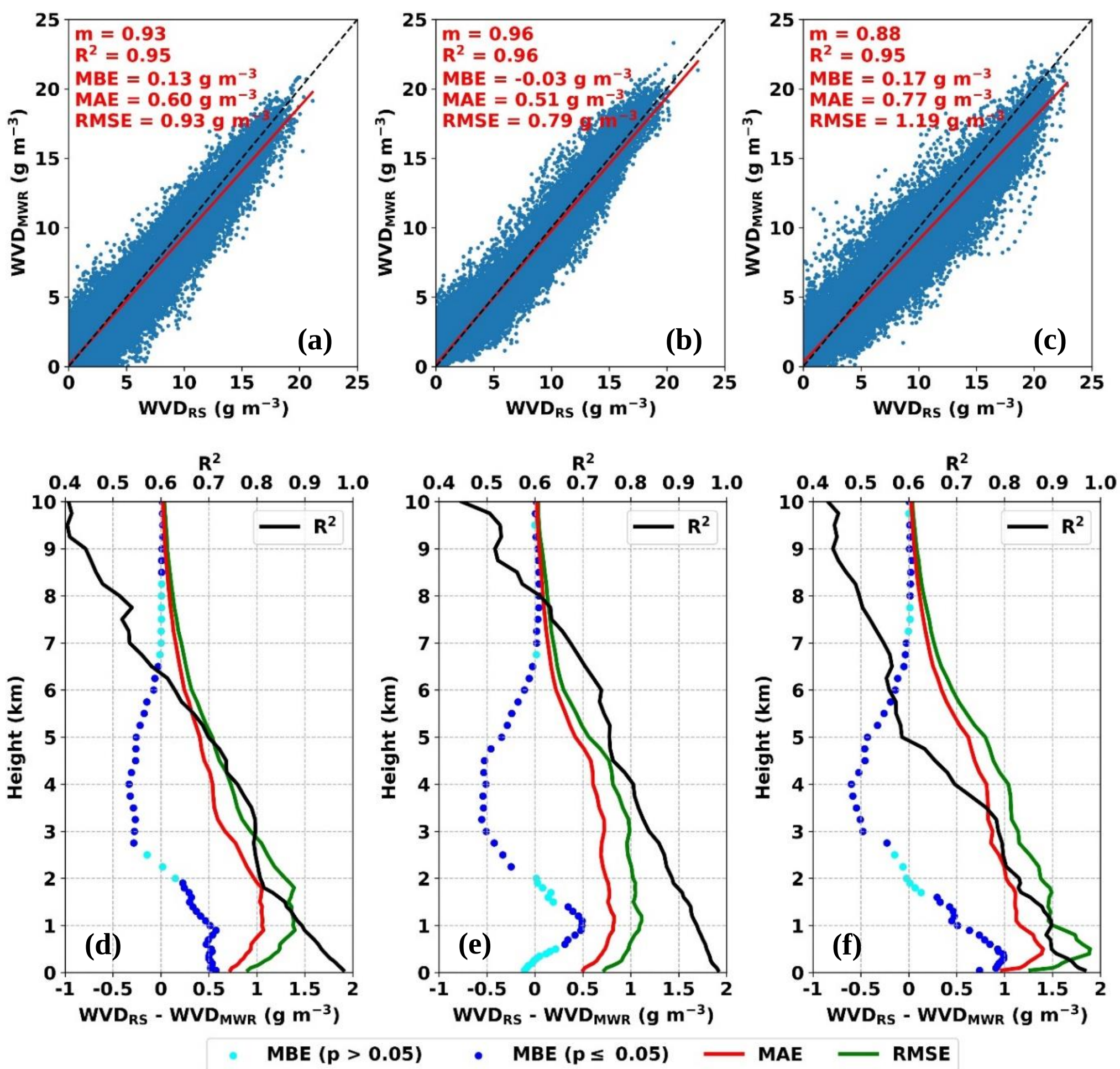

Figure 8. Scatterplots for RS and MWR measured water vapor density (WVD) at three NYSM Profiler Network sites at: (a) Buffalo, (b) Albany, and (c) Stony Brook. Vertical profiles of $R^2$, MBE, MAE and RMSE for the same variable at the respective sites: (d) Buffalo, (e) Albany, and (f) Stony Brook.

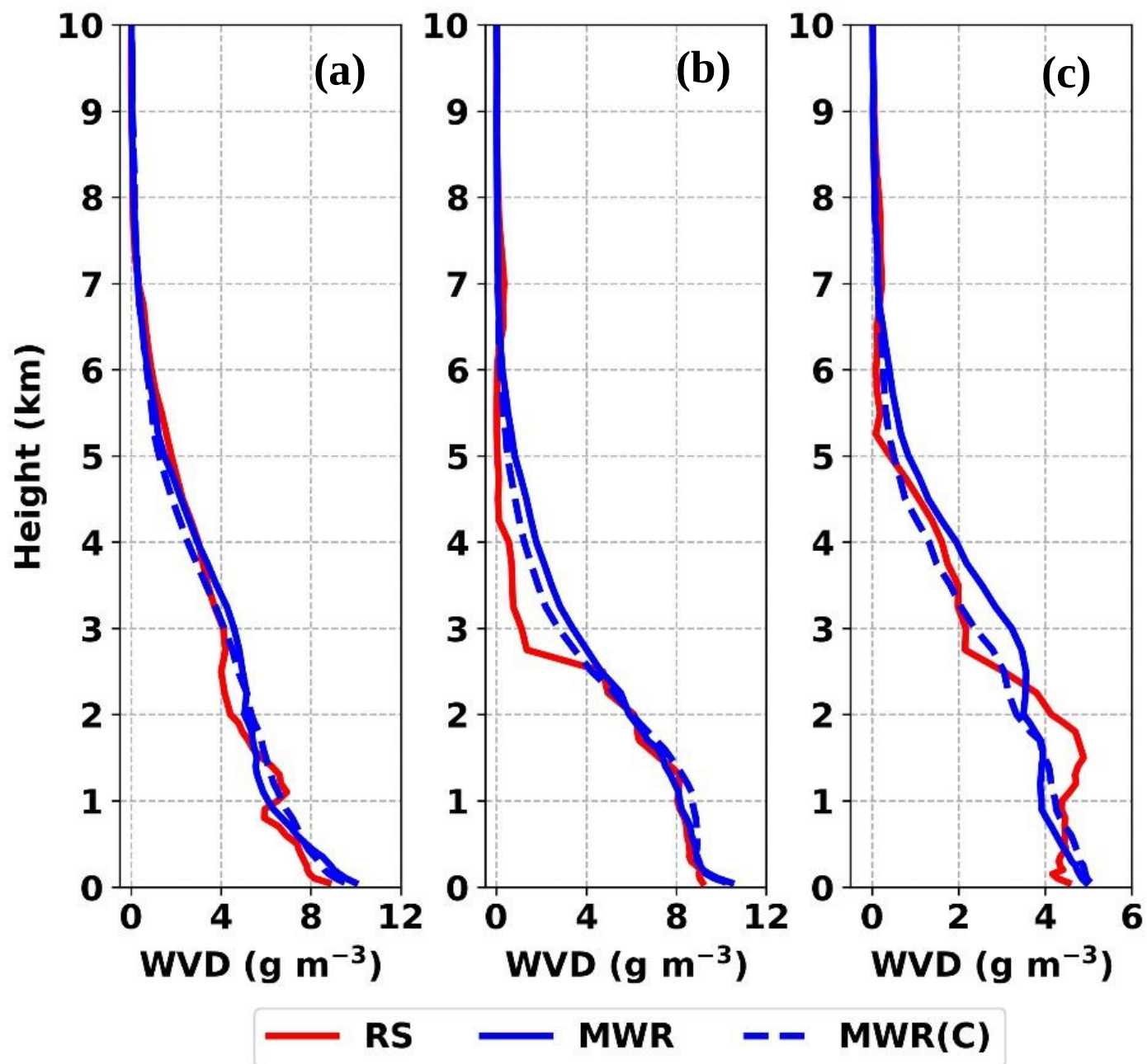

390 Figure 9. Vertical profiles of water vapor density (WVD) measured by RS and MWR – Original and Corrected (C) at (a) 23 UTC on 2 May (precipitation day), (b) 23 UTC on 28 July (cloudy day), (c) 11 UTC on 2 May 2021 (clear sky day) at Albany.

In summary, the RS and MWR measured temperature and water vapor density are strongly correlated across the three sites with $R^2$ only varying by 1% from one site to another. Both temperature and water vapor density biases are found to be statistically significant. Temperature comparisons are found to be in better agreement at lower altitudes than at higher altitudes. This is likely because the MWR measured V-band observations are ingested into the neural network with greater weighting function at lower heights than at higher heights to produce a finer vertical resolution at lower heights. In contrast, the water vapor density comparisons show better results at higher altitudes than at lower altitudes. This may be because of the highly variable moisture field within the boundary layer. The observed errors between the RS and MWR data are likely due to error inherent to the MWR neural network retrieval method and large RS drift distances as discussed in Section 3. Nevertheless, the MWR exhibits consistent behavior across three sites with similar site-to-site error statistics. This consistent performance of the MWR shows the robustness

of the instrument at across different weather and geographical locations. A summary of comparison statistics for the MWR data are presented in Table 4.

Table 4. Comparison statistics between Radiosondes and MWR data based on weather condition from January 2018 to August 2021.

| Variable | Weather | Site | m | $R^2$ | MBE | MAE | RMSE |
|---|---|---|---|---|---|---|---|
| T (°C) | All | Buffalo | 1.00 | 0.98 | 2.7 | 3.0 | 3.8 |
| | | Albany | 1.02 | 0.97 | 3.3 | 3.7 | 4.8 |
| | | Stony Brook | 1.03 | 0.98 | 2.9 | 3.4 | 4.3 |
| | Precipitation | Buffalo | 0.97 | 0.99 | 1.6 | 2.2 | 2.8 |
| | | Albany | 0.99 | 0.98 | 1.7 | 2.3 | 3.1 |
| | | Stony Brook | 1.00 | 0.98 | 2.0 | 2.6 | 3.4 |
| | Cloudy | Buffalo | 1.00 | 0.98 | 2.8 | 3.2 | 3.9 |
| | | Albany | 1.01 | 0.97 | 3.0 | 3.4 | 4.4 |
| | | Stony Brook | 1.03 | 0.98 | 2.6 | 3.1 | 3.9 |
| | Clear | Buffalo | 1.02 | 0.98 | 2.8 | 3.1 | 3.8 |
| | | Albany | 1.04 | 0.97 | 4.4 | 4.7 | 5.8 |
| | | Stony Brook | 1.02 | 0.97 | 3.7 | 4.1 | 5.1 |
| WVD (g $m^{-3}$) | All | Buffalo | 0.93 | 0.95 | 0.13 | 0.60 | 0.93 |
| | | Albany | 0.96 | 0.96 | -0.03 | 0.51 | 0.79 |
| | | Stony Brook | 0.88 | 0.95 | 0.17 | 0.77 | 1.19 |
| | Precipitation | Buffalo | 0.98 | 0.96 | 0.10 | 0.55 | 0.85 |
| | | Albany | 1.01 | 0.97 | -0.05 | 0.54 | 0.82 |
| | | Stony Brook | 0.90 | 0.95 | 0.27 | 0.81 | 1.22 |
| | Cloudy | Buffalo | 0.93 | 0.95 | 0.14 | 0.62 | 0.95 |
| | | Albany | 0.96 | 0.96 | -0.02 | 0.54 | 0.83 |
| | | Stony Brook | 0.88 | 0.95 | 0.21 | 0.86 | 1.29 |
| | Clear | Buffalo | 0.88 | 0.91 | 0.12 | 0.59 | 0.94 |
| | | Albany | 0.92 | 0.95 | -0.03 | 0.45 | 0.71 |
| | | Stony Brook | 0.85 | 0.91 | 0.06 | 0.63 | 1.01 |

### 4.4 Evaluation of MWR data collected during different weather conditions

Since the MWR is designed to perform during all types of weather conditions, the accuracy of the MWR data is analyzed separately for precipitation, cloudy and clear sky days. The MWR is equipped with a precipitation sensor that detects any precipitation over the MWR radome and provides a status flag of 0 = no precipitation and 1 = precipitation. The MWR is also equipped with an infrared radiation thermometer (IRT) that measures the cloud base temperature. The cloud base height (CBH) is set to the lowest height where the cloud base temperature is equal to the retrieved temperature profile (Ware et al., 2003). Therefore, a CBH > 0 represents a cloudy condition, CBH = -1 represents clear sky conditions, and CBH = 0 represents fog or precipitation. A total of 234, 280 and 330 profiles were selected for precipitation days at Buffalo, Albany, and

Stony Brook. Similarly, 1305, 1272 and 790 profiles were selected for cloudy days while 472, 808 and 635 profiles were selected for clear sky days at the respective sites. The overall statistical results between the RS and MWR measured temperature and water vapor density under precipitation, cloudy and clear sky days are presented in Table 4.

Temperature comparisons show high correlation with $R^2 \geq 0.97$ and are within 1% when compared across different weather conditions and sites (Table 4). Precipitation days have the lowest MBE, MAE and RMSE while the clear sky days have the greatest errors (at Buffalo, clear sky and cloudy days errors are nearly identical, only differ within 0.1 ºC). Cold temperature biases are observed during all three weather conditions across all three sites and are statistically significant ($p \leq 0.05$). Clear sky day errors are greater than those from the precipitation days by 0.9 – 2.7 ºC, whereas the cloudy day errors are greater than those from the precipitation days by 0.5 – 1.3 ºC. Along the profile, errors are similar below 1 km and are mostly within 2 ºC, regardless of weather conditions (Fig. 10 a – c, as a representative only Albany site shown). Above 1 km, the errors are at their maximum but lowest on precipitation days and highest on clear sky days. The MWR temperature cold biases are clearly evident in the example profiles shown in Fig. 7 (a – c), which are much more pronounced during clear sky (Fig. 7c) than cloudy (Fig. 7b) and precipitation day (Fig. 7a). The larger cold biases during cloudy days than precipitation days are consistent with the results by Cimini et al. (2011) whereas the larger cold biases during clear sky days than the cloudy days are consistent with the results by Xu et al. (2015).

For water vapor density, precipitation days have the highest $R^2$, and clear sky days have the lowest $R^2$ (Table 4), similar to that for temperature. The $R^2$ between precipitation and cloudy days are nearly identical (within just 1%), but precipitation and clear sky days vary by 2 – 5%. The largest errors occur on cloudy days and the lowest on clear sky days, with an exception at Buffalo where the lowest errors occur on precipitation days. All weather condition errors vary within 0.1 g m$^{-3}$ at Buffalo but up to 0.28 g m$^{-3}$ between cloudy and clear sky days at Albany and Stony Brook. Larger errors are expected during cloudy/precipitation days due to the higher variability of moisture in the clouds. Under all-weather conditions, dry biases are observed at Buffalo and Stony Brook that are statistically significant ($p \leq 0.05$) whereas a low wet biases are observed at Albany that are statistically insignificant. The error profiles for water vapor density during precipitation and cloudy days show similar values and are relatively higher than those observed during clear sky days (Fig. 10 d – f). The relatively lower errors in the MWR water vapor density during clear sky

than cloudy days are also reported in Xu et al. (2015). The MWR measured water vapor density profiles are smooth and lack high resolution vertical details regardless of the weather conditions (Fig. 9 a – c).

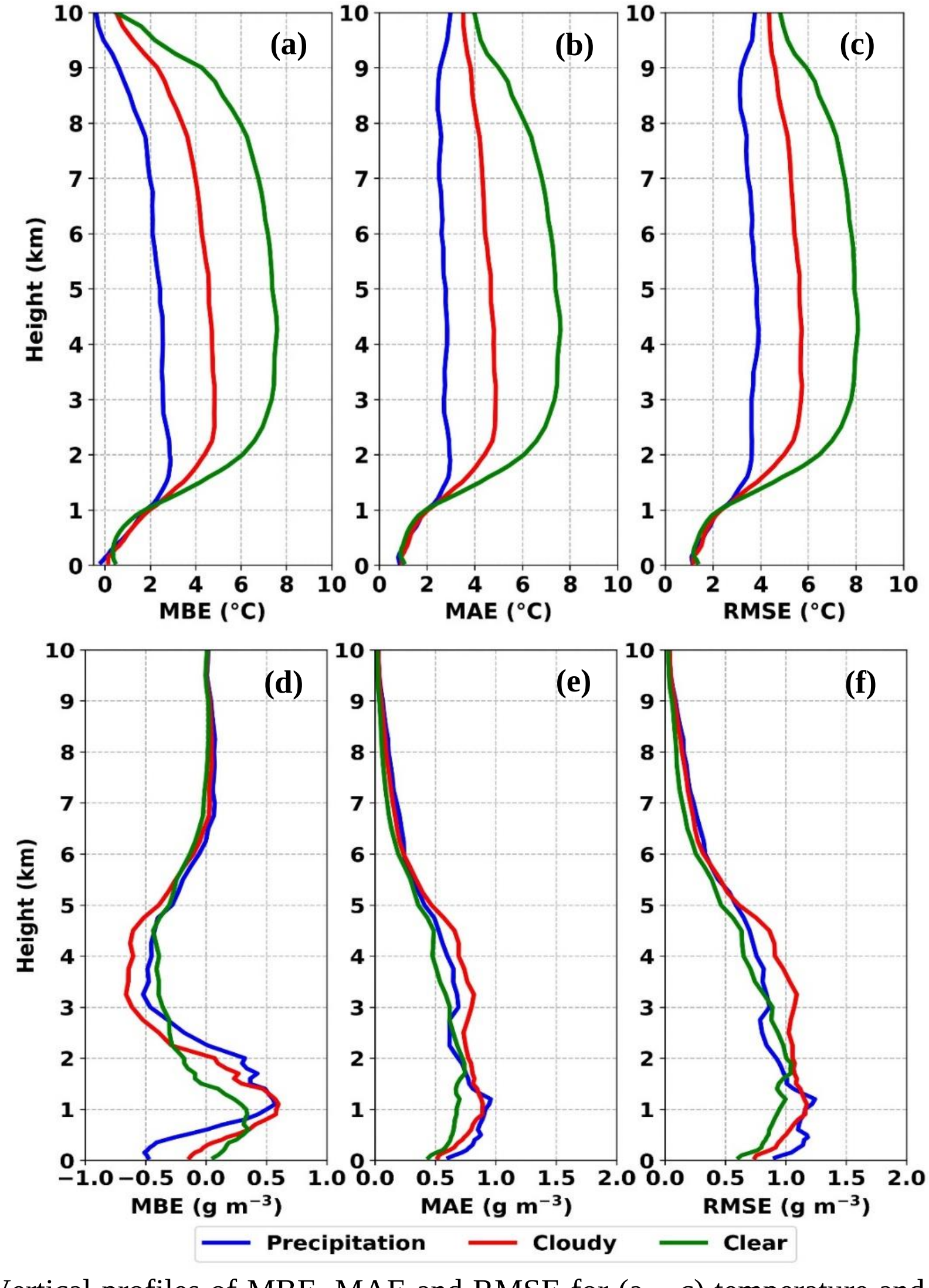

Figure 10. Vertical profiles of MBE, MAE and RMSE for (a – c) temperature and (d – f) water vapor density during three weather conditions from the NYSM Profiler site at Albany.

In summary, the MWR is found have varying performance under different weather conditions, particularly above 1 km for the temperature and below ~5km for the water vapor

density. Overall, the temperature (water vapor density) errors are largest during clear sky (precipitation/cloudy) days.

### 4.5 Correction to MWR biases

A simple correction method is developed and applied to the MWR data to minimize the biases in MWR measurements as noted in Sections 4.3 and 4.4. This method utilizes a linear regression fit as a function of height and is calculated and applied separately for temperature and water vapor density during precipitation, cloudy and clear sky days. A best fit linear model is developed at each height and for each variable. From the available profiles collected January 2018 to August 2021, 75% of profiles were randomly selected as a training dataset and the remaining 25% were used for testing and evaluation. A 10-fold cross-validation process was performed on the profiles training dataset at each height. The mean statistics from the cross-validation were then used to develop the best fit linear model. The model was then applied to correct the MWR data from the testing datasets and compared against the RS data.

The error statistics between the RS and MWR data, both original and corrected (C), during three weather conditions are presented in Fig. 11 (a – f, as a representative only Albany site shown). Error is minimized at each height during all three weather conditions. For temperature, MBE(C) is close to 0 and both MAE(C) and RMSE(C) decrease significantly for all three weather conditions (Fig. 11 a – c). Unlike MAE and RMSE, both MAE(C) and RMSE(C) increase monotonically with height, although absolute values were much more improved. As with temperature, the MBE(C) profiles for water vapor density showed significant improvement with the correction for all three weather conditions (Fig. 11 d); however, the MAE(C) and RMSE(C) showed little improvement with height (Fig. 11 e – f), which again could be due to the fact that the MWR measured water vapor density profiles are smooth and lack the vertical details that the RS is able to capture with its higher vertical resolution (examples shown in Fig. 9). In summary, this simple linear regression correction method helps to reduce systematic biases in the MWR data, which is much more pronounced in temperature than the water vapor density profiles. This is evident through the corrected individual profiles shown in Fig. 7 (temperature) and Fig. 9 (water vapor density).

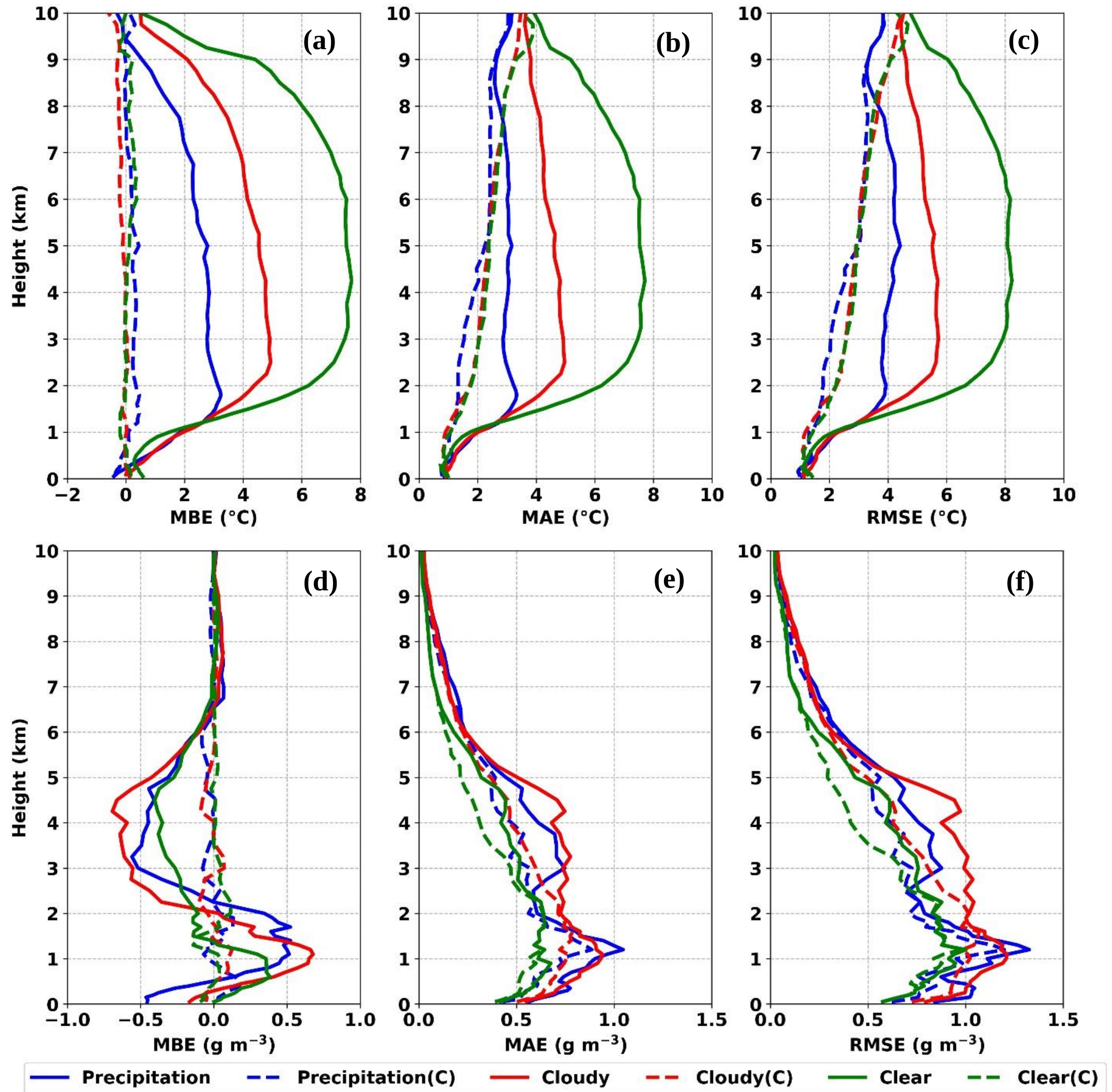

Figure 11. Vertical profiles of MBE, MAE and RMSE for original and corrected (C) MWR measured (a – c) temperature and (d – f) water vapor density during three weather conditions from the NYSM Profiler site at Albany.

### 4.6 Evaluation of MWR derived thermodynamic indices

In this section, thermodynamic indices derived from the RS and MWR are examined. For this evaluation, only corrected MWR profiles from the selected testing dataset from Section 4.5 were used to compute the 14 independent thermodynamic parameters listed in Section 3. The corrected profiles significantly reduce biases of the parameters that are mostly statistically

insignificant. On average, $R^2$ increases by 7% and MAE and RMSE decrease by 21%. The comparison results presented in Table 5 show all the MWR derived corrected parameters in good agreement with those derived from the RS with $R^2 \geq 0.55$. Except for TT (all sites), meanRH (ps – 700 hPa, both at Buffalo) and ($\theta_{es} - \theta_e$) at 850 hPa (Stony Brook), all other parameters show $R^2 \geq 0.70$. The TPW ($R^2 = 0.99$), THTK (ps – 850 hPa, $R^2 \geq 0.97$) and THTK (ps – 500 hPa, $R^2 \geq 0.93$) are the highest correlated parameters. Among the four single-level indices (KI, LI, SI and TT), LI shows the best results with the highest $R^2 \geq 0.90$ and the lowest MBE, MAE, and RMSE (≤ 3.0 ºC) across the three sites. While $R^2$ for TT is the worst among the four single-level indices, the MAE and RMSE are the highest for KI. The biases for the derived parameters are mostly statistically insignificant ($p > 0.05$) except for meanRH (ps – 700 hPa), ($\theta_{es} - \theta_e$) at 850 hPa and THTK (ps – 850 and ps – 500 hPa).

Overall, thermodynamic indices derived from the RS and corrected MWR are well correlated. Thus, the real-time forecasting parameters obtained from the MWR can be a valuable tool to forecasters during high-impact weather events, which is otherwise not possible with a typical twice daily radiosonde.

Table 5. Comparison statistics of thermodynamic parameters between Radiosonde and MWR.

| Parameter | Site | m | $R^2$ | MBE | MAE | RMSE |
|---|---|---|---|---|---|---|
| MeanRH (ps – 950hPa) (%) | Buffalo | 1.00 | 0.83 | -0.2 | 5.7 | 7.2 |
| | Albany | 0.86 | 0.85 | -0.1 | 5.3 | 6.6 |
| | Stony Brook | 0.85 | 0.84 | 0.4 | 6.1 | 7.7 |
| MeanRH (ps – 850hPa) (%) | Buffalo | 0.93 | 0.70 | 0.8 | 8.5 | 10.5 |
| | Albany | 0.76 | 0.79 | 1.9 | 6.8 | 8.4 |
| | Stony Brook | 0.81 | 0.81 | 1.6 | 6.8 | 8.6 |
| MeanRH (ps – 700 hPa) (%) | Buffalo | 0.82 | 0.68 | 3.3 | 9.1 | 11.8 |
| | Albany | 0.77 | 0.78 | 5.5 | 8.3 | 10.8 |
| | Stony Brook | 0.83 | 0.87 | 4.1 | 7.0 | 8.7 |
| TPW (inch) | Buffalo | 1.04 | 0.99 | 0.00 | 0.05 | 0.06 |
| | Albany | 1.05 | 0.99 | 0.00 | 0.05 | 0.07 |
| | Stony Brook | 0.97 | 0.99 | 0.00 | 0.05 | 0.07 |
| θ LR (ps – 850 hPa) (K km$^{-1}$) | Buffalo | 0.73 | 0.77 | -0.1 | 1.2 | 1.5 |
| | Albany | 0.70 | 0.79 | -0.1 | 1.3 | 1.6 |
| | Stony Brook | 0.82 | 0.76 | 0.0 | 1.1 | 1.3 |
| θ LR (ps – 700 hPa) (K km$^{-1}$) | Buffalo | 0.81 | 0.82 | 0.0 | 0.7 | 0.8 |
| | Albany | 0.76 | 0.81 | -0.1 | 0.8 | 0.9 |
| | Stony Brook | 0.75 | 0.76 | -0.1 | 0.7 | 0.8 |
| $\theta_{es}$ - $\theta_e$ (950 hPa) (K) | Buffalo | 0.98 | 0.93 | -0.3 | 1.9 | 2.5 |
| | Albany | 0.93 | 0.91 | -0.3 | 1.8 | 2.5 |
| | Stony Brook | 0.80 | 0.81 | -0.3 | 2.8 | 3.5 |
| $\theta_{es}$ - $\theta_e$ (850 hPa) (K) | Buffalo | 0.84 | 0.77 | -0.8 | 2.8 | 3.5 |
| | Albany | 0.67 | 0.70 | -0.7 | 2.5 | 3.1 |
| | Stony Brook | 0.63 | 0.61 | -1.0 | 3.2 | 3.9 |
| THTK (ps – 850 hPa) (km) | Buffalo | 0.98 | 0.98 | 0.01 | 0.01 | 0.02 |
| | Albany | 0.98 | 0.98 | 0.01 | 0.01 | 0.02 |
| | Stony Brook | 0.98 | 0.97 | 0.02 | 0.02 | 0.02 |
| THTK (ps – 500 hPa) (km) | Buffalo | 0.87 | 0.93 | 0.02 | 0.05 | 0.06 |
| | Albany | 0.97 | 0.93 | 0.04 | 0.05 | 0.07 |
| | Stony Brook | 0.96 | 0.94 | 0.05 | 0.05 | 0.06 |
| KI (°C) | Buffalo | 0.81 | 0.77 | 0.8 | 7.7 | 9.8 |
| | Albany | 0.81 | 0.80 | 1.9 | 8.4 | 10.4 |
| | Stony Brook | 0.77 | 0.81 | 1.0 | 8.0 | 10.0 |
| LI (°C) | Buffalo | 0.93 | 0.90 | -0.1 | 2.4 | 2.9 |
| | Albany | 0.89 | 0.91 | -0.1 | 2.4 | 3.0 |
| | Stony Brook | 0.90 | 0.90 | 0.2 | 2.3 | 2.9 |
| SI (°C) | Buffalo | 0.85 | 0.80 | -0.2 | 2.6 | 3.2 |
| | Albany | 0.79 | 0.80 | -0.2 | 2.7 | 3.4 |
| | Stony Brook | 0.77 | 0.79 | -0.2 | 2.5 | 3.2 |
| TT (°C) | Buffalo | 0.67 | 0.55 | 0.3 | 5.8 | 7.4 |
| | Albany | 0.66 | 0.65 | 0.5 | 5.6 | 7.1 |
| | Stony Brook | 0.62 | 0.66 | 0.0 | 5.1 | 6.6 |

### 4.7 A case study of a thunderstorm event

A thunderstorm event is examined using the thermodynamic and wind shear parameters derived from the MWR and DL. On 12 August 2021, the National Weather Service reported a severe thunderstorm at Albany from 14:40 to 15:30 LT with heavy rainfall of 1.04 in/hr and maximum wind gust of 60 mph. Figure 12 shows the temporal variations of temperature, vapor density, liquid density, and relative humidity from the MWR and CNR from the DL overlaid with wind barbs from 9 to 19 LT. A sharp increase in vapor density between 1000 and 800 hPa (Fig. 12b), liquid density between 900 and 600 hPa (Fig. 12c) and relative humidity up to 500 hPa (Fig. 12d) are observed shortly after 14 LT. Similarly, the wind speed within the lowest 1 km AGL doubles (10 – 15 knots to 25+ knots) from 14 to 15 LT (Fig. 12e) with a change in wind direction from southerly/southwesterly to mostly northwesterly.

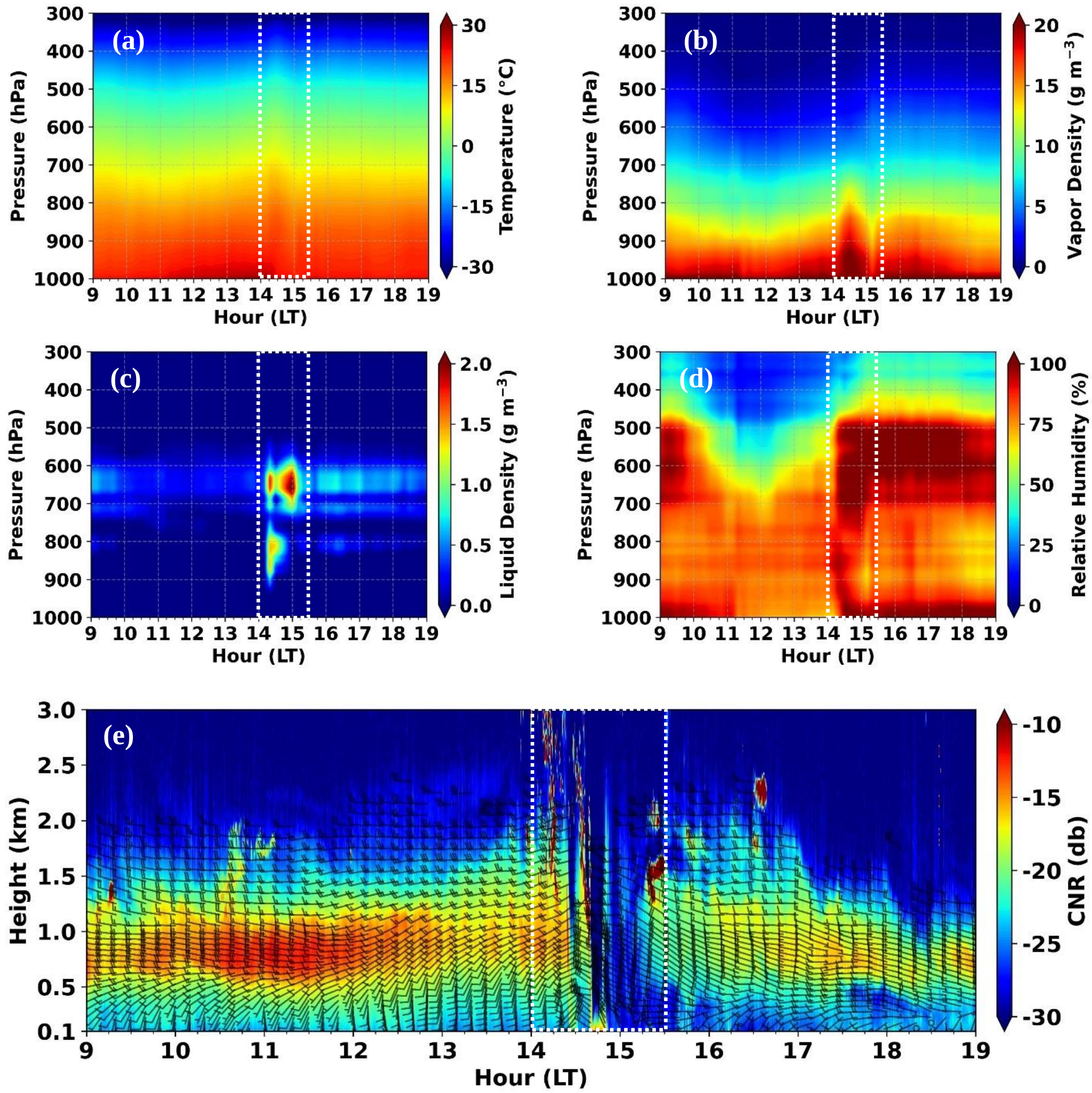

Figure 12. The MWR measured time-height cross section plots for (a) temperature (ºC), (b) vapor density (g $m^{-3}$), (c) liquid density (g $m^{-3}$), (d) relative humidity (%) and (e) DL measured CNR (dB) with 10 minutes averaged wind barbs at Albany on 12 August 2021. Dotted box represents thunderstorm episode.

Figure 13 (a – h) shows distinctive temporal variations before and during the storm. The dew point temperature (DWPT) at 850 hPa slowly increases from 11 LT while the DWPT at 1000 hPa starts to increase an hour later around 12 LT, both increasing by ≥ 3 ºC within 30 minutes of reaching peak at 14:20 and 14:30 LT respectively, just prior to thunderstorm genesis (Fig. 13a).

The TPW decreases until 12 LT but then starts to increase and reaches a peak at 14:30 LT just before storm initiation (Fig. 13b). Both DWPT and TPW increase for 2 – 3 hours with a sharp increase ~30 minutes prior to the storm. All three levels of mean RH (1000 to 950, 850 and 700 hPa) increase sharply starting at 14 LT (coincident with the sharp increase in DWPT and TPW) and reach ≥ 90%, just prior to the storm (Fig. 13c). Both potential temperature (θ) LRs (1000 – 850 hPa and 1000 – 700 hPa) decrease continuously until 14 LT, indicative of instability prior to the thunderstorm occurrence (Fig. 13d). The stability index ($\theta_{es} - \theta_e$) at two levels (950 and 850 hPa) decrease sharply from 14 LT reaching the minimum value at 14:40 just before the storm initiation, suggesting a change from the warmer unsaturated to cooler saturated atmosphere (Fig. 13e). A KI ≥ 30 °C indicates a moderate chance for thunderstorms with rain while the KI ≥ 40 °C indicates a high chance for thunderstorms with heavy rain. There is a relative increasing trend in the KI after 12 LT (~2.5 hours prior to the storm) where the KI increases roughly by 5 °C between 12 and 13:40 LT and further increases by ~10 °C in 40 minutes reaching the peak value of 44.8 °C at 14:30 LT (Fig. 13f, blue line). A TT ≥ 45 °C indicates the possibility of thunderstorms while TT ≥ 50 °C indicates a possibility of severe thunderstorms. The TT values are >45 °C from 10:20 LT through the end of the storm event (Fig. 13f, red line). From 13:50 to 14:20, TT increases by >4 °C and reaches the peak value of 49.6 °C just prior to thunderstorm genesis. The more negative an LI and SI, the greater the instability. LI is mostly between 0 and -3 °C until 14:10 LT, drops below -3 °C and reaches minimum values of -4.3 °C at 14:30 LT (Fig 14g, blue line). SI drops steadily until 13:30 LT and then drops precipitously below – 3 °C between 14:10 and 14:40 LT (Fig 13g, red line). Finally, the wind shear (100 m – 1 km) is mostly < 4 m $s^{-1}$ (~8 knots) until 14:50 LT and then drastically increases to 12 m $s^{-1}$ (~23 knots) at 15 LT, shortly after the thunderstorm begins (Fig. 13h). Such a significant increase in shear generally indicates increasing storm severity. In summary, using a combination of one or more convective index parameters from a collocated DL and MWR, it’s possible to monitor low-level moisture, instability, and wind shear for storm initiation and severity. With the normal radiosonde launch times (00 and 12 UTC) outside of this 10-hour window, crucial details of the thunderstorm could have been easily missed without the NYSM Profiler Network.

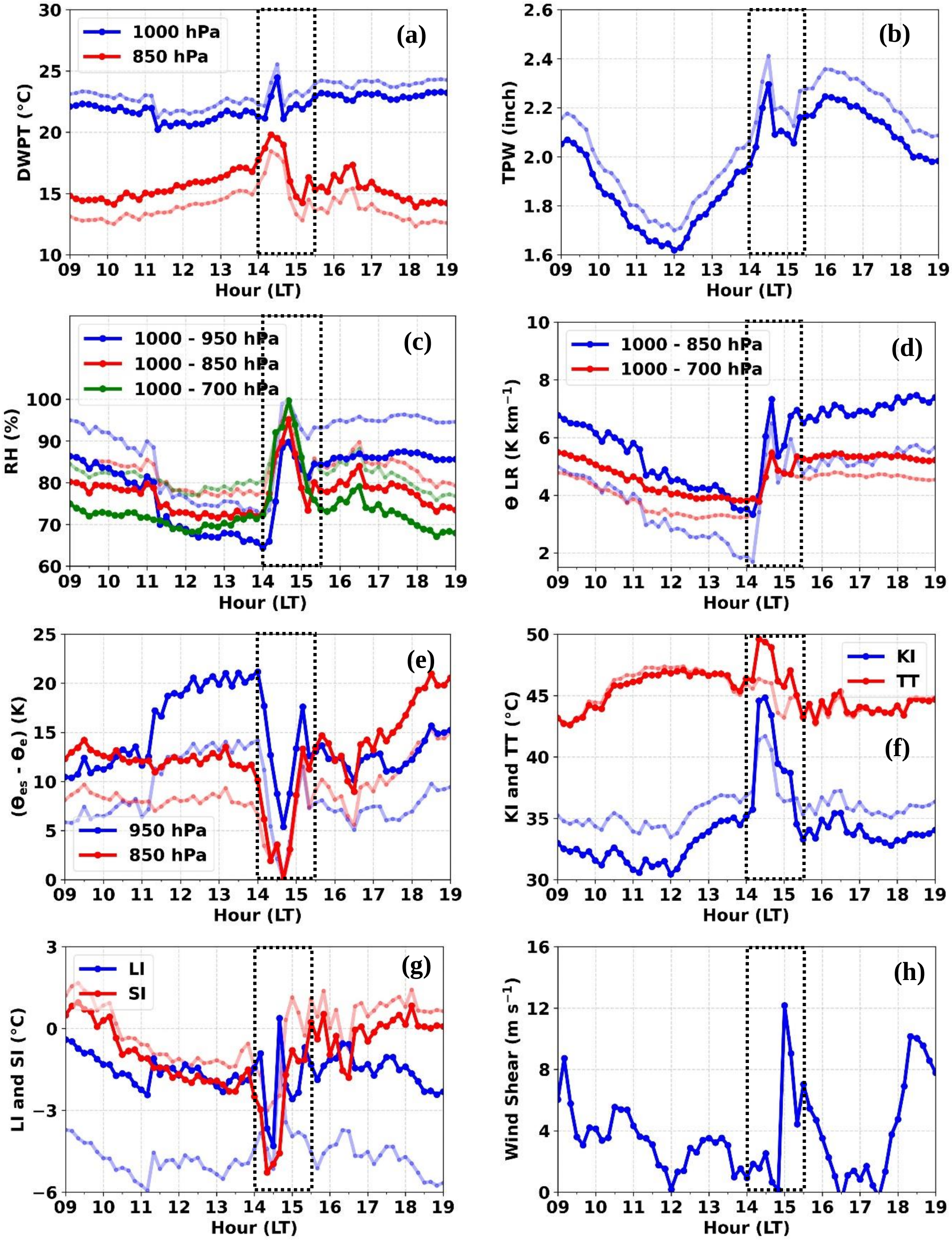

Figure 13. Original (light color) and corrected (bold color) MWR derived (a) DWPT, (b) TPW, (c) Mean RH (d) θ LR (e) ($\theta_{es}$ - $\theta_e$), (f) KI and TT, (g) LI and SI, and DL derived (h) Wind Shear (100 m – 1 km) at Albany on 12 August 2021. Dotted box represents thunderstorm episode.

### 4.8 Evaluation of eSIR AOD Data

Measurements of aerosol optical depth (AOD) as computed by the NYSM Profiler eSIR and AERONET were compared at Stony Brook and Bronx (Fig. 14 a – f). The highest $R^2$ observed for AOD was at 500 nm wavelength ($R^2 \geq 0.92$) and the lowest $R^2$ observed was at 1040 nm wavelength ($R^2 \geq 0.78$) at both sites. Discrepancies at the 1040 nm wavelength could be due to the influence of trace gases such as $CO_2$, $O_2$, CO, NOx, $CH_4$ and $SO_2$. The eSIR-derived AOD only considers the optical depth contribution from Rayleigh scattering, water vapor and ozone. The errors are within the expected range based on the accuracies of the eSIR and AERONET measurements listed in Table 1. The AOD biases are found to be statistically significant ($p < 0.05$) except for 1020 nm AOD at Bronx.

In summary, AOD estimates from the eSIR and AERONET show close agreement with each other at both sites with AOD measurements at lower wavelengths comparatively better than at higher wavelengths. Having accurate AOD data is valuable for air quality studies and forecasting because of its frequent use in the estimation of surface $PM_{2.5}$ (Kumar et al., 2007; Schaap et al., 2009; Chudnovsky et al., 2014; Xie et al., 2015) and the classification and characterization of aerosol types and size (Eck et al., 1999; Schuster et al., 2006; Khan et al., 2016; Torres and Fuertes, 2021).

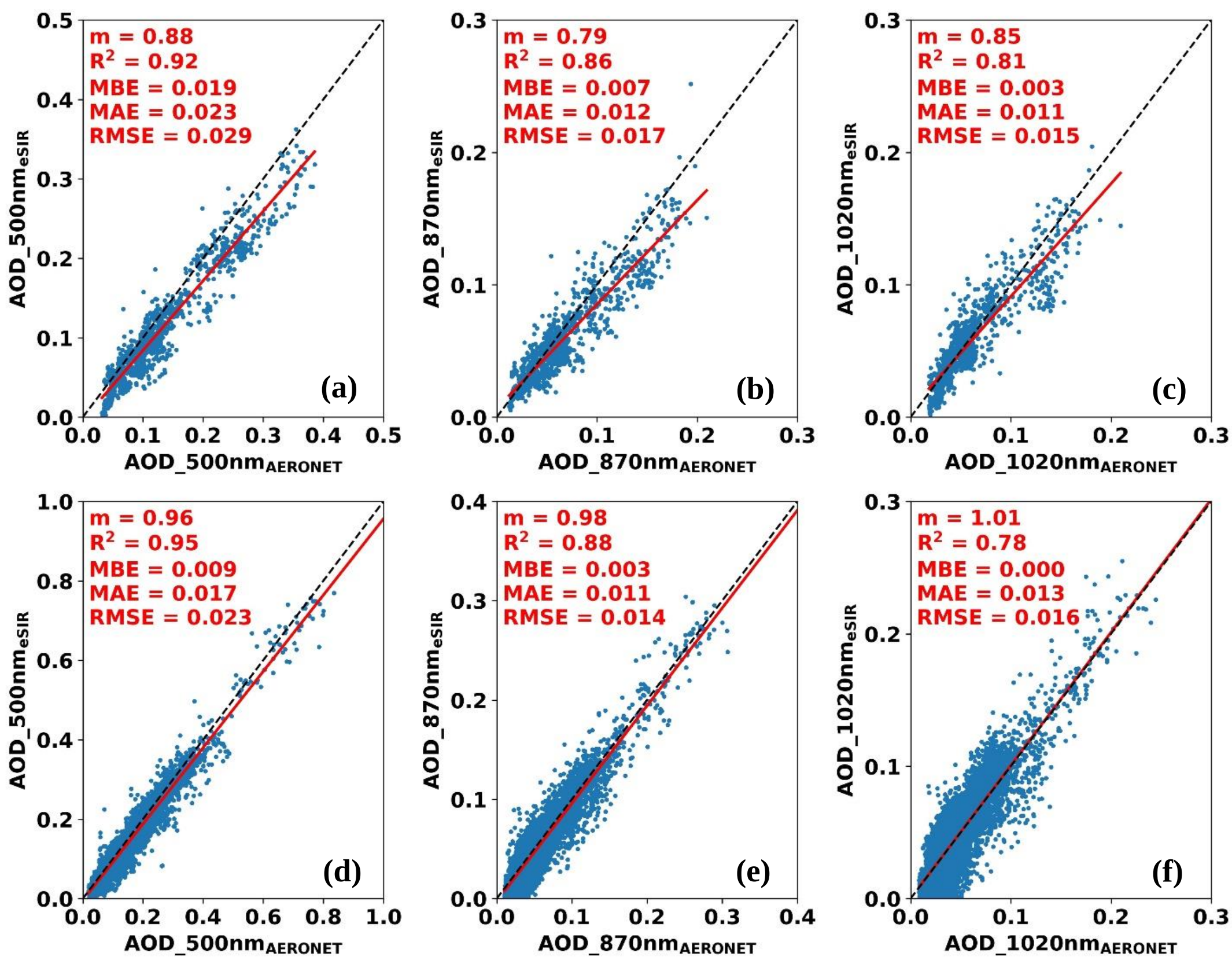

Figure 14. Scatterplots for the eSIR and Aeronet derived AOD for three channels: 500 nm, 870 nm, and 1020 nm at (a – c) Stony Brook and (d – f) Bronx.

## 5 Summary and conclusions

The primary objective of this study is to compare and assess the NYSM Profiler Network data with respect to in situ reference measurements from the NWS radiosondes and AERONET. Data from January 2018 to August 2021 were used to assess the accuracy of wind speed up to 3 km from the DL and temperature and vapor density up to 10 km from the MWR. These data were evaluated at three NYSM Profiler Network sites (Buffalo, Albany, and Stony Brook) against radiosonde measurements. Similarly, data from April to June 2018 and from March 2018 to October 2019 were used to assess the accuracy of AOD derived from the eSIR at Stony Brook and Bronx with respect to AERONET AOD measurements.

The comparison results show $R^2 \geq 0.89$ for wind speed and $R^2$ mostly exceeding 0.86 for wind shear (100 m – 1 km and 100 m – 3 km) measurements with MAE and RMSE below 2.5 m

$s^{-1}$ across the three sites. MBE is found to be statistically equal to 0 nearly at all height levels. Wind speed measurements above 0.5 km are found to be better correlated than below 0.5 km due to irregular radiosonde motions in the near surface turbulent layer. Site-to-site MAE and RMSE differences for both wind speed and wind shear are ≤ 0.7 m $s^{-1}$, indicating consistent performance of the DL across multiple sites.

The estimates of temperature and water vapor density from the MWR and RS show an overall high correlation of $R^2 \geq 0.95$ across the three sites. The MAE and RMSE for temperature and water vapor density are below 4.8 ºC and 1.19 g $m^{-3}$ respectively, with MBE statistically significant and different from 0. The temperature errors are found to be lower within the boundary layer than above it while the water vapor density shows the opposite trend. Overall site-to-site MAE and RMSE differences for temperature and water vapor density are ≤ 1.0 ºC and 0.4 g $m^{-3}$, respectively. The relatively small differences and similar vertical structure in error profiles for the MWR data demonstrate a consistent performance of the MWR across the different geographical sites. This also implies that the existence of discrepancies between MWR and RS data are primarily due to the MWR itself, likely related to the neural network retrieval technique since two different types of radiosondes are used at Buffalo/Albany and Upton. The overall statistics of the MWR data evaluated in different weather conditions (precipitation, cloudy and clear sky days) show somewhat varying performance. Correlations are found to be best (worst) during the precipitation (clear sky) days. Similarly, temperature errors are smaller (larger) on precipitation (clear sky) days whereas the water vapor density errors are relatively smaller (larger) errors on clear sky (cloudy/precipitation) days. Because of a consistent bias observed in the MWR data with reference to the RS data, a linear bias correction is developed and applied. This method reduces the systematic biases significantly with improvement in temperature much more pronounced than water vapor density. Finally, the corrected MWR data are used to retrieve 14 different thermodynamics parameters and are compared against those derived from the RS data. All 14 parameters have $R^2 \geq 0.55$ across the three sites. Except for TT (all three sites) and meanRH (ps – 700 hPa at Buffalo) and ($\theta_{es} - \theta_e$) at 850 hPa (Stony Brook), all other parameters have $R^2 \geq 0.70$, which demonstrate a value and reliability of the MWR for use in the monitoring of severe convection. Most of these parameters have no statistical bias. Overall, the MWR is a robust and reliable tool for the continuous measurement of atmospheric data and derived forecasting parameters.

Finally, AODs as measured by the eSIR and AERONET show high correlations at both sites (Stony Brook and Bronx). The AOD comparisons for 500 nm wavelength show $R^2 \geq 0.92$, whereas the $R^2 \geq 0.78$ for the 1020 nm wavelengths. Similar error statistics between the eSIRs at the two sites demonstrates a consistent performance.

A profiling station, consisting of a DL, MWR, and eSIR, provides a means for continuous monitoring of the lower boundary layer winds, aerosols, thermodynamic variables, spectral direct and diffuse radiations at high resolutions. A network of such stations allows for regional monitoring, spatial comparisons, and neighborhood checks for quality control, ensuring a more accurate analysis. Overall, the NYSM Profiler Network provides low-level atmospheric and aerosol optical data with relatively high accuracy. While some temperature and moisture biases are found with the MWR, these errors can be corrected with a simple linear fit. A multi-year, multi-station evaluation of the NYSM Profiler Network sensors show minimal differences across different sites and meteorological conditions. As demonstrated, such a network can be useful for improving situational awareness during high-impact weather operations with its timely and much improved spatial and temporal monitoring of the boundary layer.

*Data Availability Statement.* The NYSM Profiler Network data is available at http://www.nysmesonet.org/weather/requestdata according to the NYSM data policy stated in the webpage. The NWS radiosonde data is available at the University of Wyoming Atmospheric Science Radiosonde Archive, http://weather.uwyo.edu/upperair/bufrraob.shtml and the AERONET data is available at https://aeronet.gsfc.nasa.gov/new_web/aerosols.html.

*Author contributions.* BS designed the study, analyzed the results, and prepared the original manuscript. JB and JW reviewed, suggested, and edited the manuscript.

*Competing interests*. The authors declare that they have no conflicts of interest.

*Acknowledgements.* This research is made possible by the New York State (NYS) Mesonet and its dedicated staff, especially our profiler technician Steven Perez for maintaining the profiler network. Original funding for the NYS Mesonet was provided by Federal Emergency Management Agency grant FEMA-4085-DR-NY, with the continued support of the Research Foundation for the State University of New York (SUNY); the University at Albany, SUNY; the Atmospheric Sciences Research Center (ASRC) at SUNY Albany; and the Department of Atmospheric and

Environmental Sciences (DAES) at SUNY Albany. This work supported is partially supported by the Observations Program within the NOAA/OAR Weather Program Office under Award No. NA21OAR4590376. This project was also supported in part by the National Mesonet Program. Special thanks to the University of Wyoming, Department of Atmospheric Science for NWS Radiosonde data and AERONET for AOD data.

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
