# Peer review of "Evaluation of the New York State Mesonet Profiler Network Data Bhupal Shrestha1, Jerald A. Brotzge2 and Junhong Wang1,3"

_Atmospheric Measurement Techniques, 2022_

## Referee Comment (RC1)

**amt-2022-85: Evaluation of the New York State Mesonet Profiler Network Data**

**Bhupal Shrestha, Jerald A. Brotzge and Junhong Wang**

The paper examines data from 3 out of 17 NYSM Profiler Network stations and analyzes wind speed from the Doppler Lidar (DL), temperature and humidity profiles from the microware radiometer (MWR), and AOD from AODeSIR. Data from the DL and MWR are evaluated against radiosondes soundings and the usefulness for prediction of severe weather events evaluated. AOD at 2 sites is evaluated against data from the Aerosol Robotic Network.

The paper is interesting, and it presents a good overview of the Network data and capabilities. I do have some general major comments on the results and related discussion. All my comments are related to the MWR evaluation that in my opinion is the most problematic part.

**Major general comments**

The purpose of the paper as stated by the authors is *"to determine the robustness and accuracies of the instruments deployed with respect to well-defined measurements"* in view of their extensive use to *"complement radiosondes and satellite systems"*. The purpose of the paper is therefore very relevant considering the need for such measurements.

With this perspective in mind, I feel that the conclusions that: *"Overall, the MWR is a robust and reliable tool for the continuous measurement of atmospheric data and derived forecasting parameters"* is not supported by the data presented. The MWR data in the paper unfortunately present a bleak outlook for those of us hoping to use microwave radiometers for temperature and humidity monitoring. In fact, the results seem to suggest that, unless radiosondes are available at the site (to help with corrections or debiasing) the RMS errors shown in this paper (Figs. 6 and 10) ranging between 2K and 8K are well beyond what is required for any purpose.

Therefore, the most important questions the authors should address in my opinion are the following:

1. Are these results due to poor calibration of the instruments and to poor retrievals? Are these issues resolvable?

2. What are the accuracy requirements we need to strive for in a ground-based network and how far are the Mesonet radiometers from these requirements? For example, in Table 1 of Wulfmeyer et al. (DOI: 10.1002/2014RG000476) a bias < 0.5 K with noise error of < 1K are reported as desirable for temperature profiling. I understand that microwave radiometers can't achieve that, but what are the accuracy requirements for the Mesonet network?

3. What are the implications of this analysis for the network itself? It seems to me that with those uncertainties only radiometers co-located with radiosondes can be reasonably used. How about the remaining 14 radiometers for which there are no radiosondes available?

This question brings the following consideration: what is the expected uncertainty in the forecast capability at a site where there are no radiosondes for correction? To this end, in my opinion, the paper would be more informative if the radiosondes were used only for evaluation purpose and the analysis was carried out entirely without the correction part. Table 5 and the whole case study (i.e. sections 4.6 and 4.7) should contain the results from the uncorrected MWR profiles. This would give us an idea of what can be reasonably expected from a profiler at a site without radiosondes.

**Major specific comments:**

1. Throughout the paper the MWR retrievals are called "measurements". Please change this. Some specific examples are reported later.

2. **Section 4.4** This section is a little bit puzzling, and I am not sure I understand it. I understand that, because the radiometer is pointing at 20-degree elevation, you may have a better chance of having less measurement degradation during rain. However, why should the radiometer do so much better during cloudy conditions than during clear sky? Or during rain than during cloudy? I assume that, if the effect of rain deposition is eliminated, the radiometer shouldn't distinguish between rainy and cloudy. To me the results could be easily explained by compensating measurement biases during precipitation, I am not sure this section should be kept at all as it is difficult to interpret.

3. **Section 4.4** How do you know what are the conditions off-zenith? The IRT is looking at zenith so how do you know whether off-zenith, where the radiometer is pointing, is cloudy or clear? Perhaps there are clouds over the radiometer and not off zenith, or the other way around. In my opinion this entire section 4.4 should be eliminated.

4. **Section 4.5** This section should be eliminated. Of course, it comes without saying, that if we do have radiosondes at the site and we correct the radiometer based on the radiosondes we get good results, we already know this. But this defeats the entire purpose of having a network of radiometers. As mentioned in my general comments, I think the manuscript would be much more valuable if the authors used the radiosondes only for evaluation and not for correction. With the correction in place the conclusion of the analysis should be that the MWR are a good "*complement*" to radiosondes and should be deployed at radiosondes sites to increase the temporal coverage but have no value at sites where there are no radiosondes.

**Minor comments:**

1. Page 6 L138 Please change vertical resolution to vertical grid
2. Page 9 Line 209 Please change MWR measurements with MWR retrievals

3. Page 9 Line 225 and following. It is important to understand that MWR measurements have no *vertically resolved information* above roughly 2 km. It will be enough to keep the MWR comparison between 0 and 3 km as well.
4. Page 10 line 230: "directly measured" please change to *retrieved* as you are comparing retrievals of temperature and humidity from the MWR with those measured by radiosondes.
5. Page 15 section 4.3: The high biases and RMSE of the MWR retrievals are obviously concerning and are probably due to poor calibration of the instruments combined to an inadequacy of the neural network retrievals. Both aspects can be improved operationally to lower the RMSE to less than 1.5 K between 0 and 3 km. Is the mesonet network planning to do that?
6. Line 363-406. The discussion of the vapor density profiles would be more informative if the author could provide the range of vapor density at the sites. This would give the reader an idea of the error percentage on the profiles (for example an RMSE of 1 $g/m^3$ would be about 10% if the average vapor density at the site is 10 $g/m^3$).

**Conclusions**

In conclusion, I think the information presented in the paper is valuable, however it would be more valuable if the authors could provide the reader with a true assessment of the value of the entire MWR network for forecast purpose. To this end the authors should not be afraid of presenting less than perfect results if that is what the MWR network is providing. Such information would be extremely valuable for those planning to deploy such networks where radiosondes are not available.
In order to evaluate the true capability of a network of radiometers for forecasting purposes the authors should use radiosondes only for evaluation and not for correction. After the uncertainties in the forecast parameters have been established (with the help of radiosondes) data from sites without radiosondes should be used to forecast events and to establish realistic uncertainties. If the results are not satisfactory the authors should discuss how the poor radiometric performance can be improved without help from radiosondes, i.e. by reviewing the calibration procedures and improving the retrievals.

---

## Author Comment (AC1)

**amt-2022-85: Evaluation of the New York State Mesonet Profiler Network Data**

**Bhupal Shrestha, Jerald A. Brotzge and Junhong Wang**

The paper examines data from 3 out of 17 NYSM Profiler Network stations and analyzes wind speed from the Doppler Lidar (DL), temperature and humidity profiles from the microware radiometer (MWR), and AOD from AODeSIR. Data from the DL and MWR are evaluated against radiosondes soundings and the usefulness for prediction of severe weather events evaluated. AOD at 2 sites is evaluated against data from the Aerosol Robotic Network.

The paper is interesting, and it presents a good overview of the Network data and capabilities. I do have some general major comments on the results and related discussion. All my comments are related to the MWR evaluation that in my opinion is the most problematic part.

**Major general comments**

The purpose of the paper as stated by the authors is *"to determine the robustness and accuracies of the instruments deployed with respect to well-defined measurements"* in view of their extensive use to "*complement radiosondes and satellite systems*". The purpose of the paper is therefore very relevant considering the need for such measurements.

With this perspective in mind, I feel that the conclusions that: "*Overall, the MWR is a robust and reliable tool for the continuous measurement of atmospheric data and derived forecasting parameters*" is not supported by the data presented. The MWR data in the paper unfortunately present a bleak outlook for those of us hoping to use microwave radiometers for temperature and humidity monitoring. In fact, the results seem to suggest that, unless radiosondes are available at the site (to help with corrections or debiasing) the RMS errors shown in this paper (Figs. 6 and 10) ranging between 2K and 8K are well beyond what is required for any purpose.

Response: We agree that the "robust and reliable" statement is not an accurate representation of the results. So, the above stated sentence is reworded to "Overall, the MWR provides continuous and real-time measurement of atmospheric data and can be a valuable nowcasting tool for high-impact weather events despite cold biases in the temperature data" in the revised manuscript (line 653-656).

Therefore, the most important questions the authors should address in my opinion are the following:

1. Are these results due to poor calibration of the instruments and to poor retrievals? Are these issues resolvable?

Response: The observed cold biases in the MWR temperature are consistent with previous studies by Cimini et al., 2011, Xu et al., 2015 and Cimini et al., 2015. These studies have attributed such cold biases to neural network, calibration uncertainty, gas absorption model,

radiosonde biases and radiosonde drift distances. However, the near identical error profiles from three selected NYSM Profiler sites suggest that those errors are not likely due to calibration issues or poor neural network performance, as those errors are sensor specific. The cold bias error has not been fully understood yet, and it is still under investigation. Though overall off-zenith (20 degrees elevation) retrievals are better than zenith retrievals (Xu et al., 2013), off-zenith retrievals have reduced MWR lower v-band weighting functions; for example, the zenith weighting function of 2 km is reduced to off-zenith weighting function of ~0.8 km. The observed vertical profile errors are consistent with such rapidly fading weighting function above ~1 km for off-zenith retrievals.

These issues are resolvable with the application of the one-dimensional variational (1-DVAR) technique (Cimini et al., 2011). While that retrieval technique is under evaluation and consideration for long-term use, the radiosonde correction method discussed in the manuscript will be implemented as a temporary solution. Since our studies have shown that limited clear sky days radiosonde data can help to reduce biases in MWR retrievals, the NYSM is planning to launch several radiosondes to bias correct remaining 14 MWRs and to further investigate the performance of the MWR during the daytime (7am – 7pm LT) apart from the twice-daily NWS launches. This information is added in line 670-672.

2. What are the accuracy requirements we need to strive for in a ground-based network and how far are the Mesonet radiometers from these requirements? For example, in Table 1 of Wulfmeyer et al. (DOI: 10.1002/2014RG000476) a bias < 0.5 K with noise error of < 1K are reported as desirable for temperature profiling. I understand that microwave radiometers can't achieve that, but what are the accuracy requirements for the Mesonet network?

Response: The accuracy requirements are set by our users of the data and vary with application. Based on WMO OSCAR (https://space.oscar.wmo.int/requirements), the "ideal" and "optimal" requirements for temperatures and humidity in PBL and free troposphere are 0.5K and 1 K, and 2% and 5% respectively, for high resolution numerical weather prediction applications. But, considering the brightness temperature accuracies and the ill-posed retrieval problem, the expected accuracies for MWR temperature and water vapor density are $0.5 – 2$ K and $0.2 – 1.5$ g m$^{-3}$ that further decrease with height (Illingworth et al., 2015). The NYSM aims to achieve similar accuracies of temperature and water vapor density, particularly within the boundary layer by minimizing any significant biases. Therefore, we presented the linear correction technique using the radiosonde data to bias correct the MWR retrievals as demonstrated in the manuscript.

Specifically, the NYSM is developed primarily with an aim to monitor and provide situational awareness and warning of severe high-impact weather events. Despite the limitations of MWR retrievals, the vertical profiles of temperature and water vapor density have shown similar trends to that from the radiosondes, particularly within the boundary layer. Thus, the high temporal resolution of vertical profiles provided by the MWR are helpful to understand the evolution of weather events and provide much needed real-time information to monitor and nowcast severe high-impact events as presented in the manuscript.

3. What are the implications of this analysis for the network itself? It seems to me that with those uncertainties only radiometers co-located with radiosondes can be reasonably used. How about the remaining 14 radiometers for which there are no radiosondes available?

This question brings the following consideration: what is the expected uncertainty in the forecast capability at a site where there are no radiosondes for correction? To this end, in my opinion, the paper would be more informative if the radiosondes were used only for evaluation purpose and the analysis was carried out entirely without the correction part. Table 5 and the whole case study (i.e. sections 4.6 and 4.7) should contain the results from the uncorrected MWR profiles. This would give us an idea of what can be reasonably expected from a profiler at a site without radiosondes.

Response: As a short-term plan, the NYSM will launch a several-day series of radiosondes near each profiler site to bias correct the MWR retrievals. Bias corrections will be developed for the remaining 14 Profiler sites to yield the accuracies discussed above in (2). Our data has shown that bias corrections are stable with time, and so a limited set of radiosondes collected at each site during clear sky days are sufficient, until a 1-DVAR technique or other new retrieval method is implemented in near future.

A height-dependent correction method applied to the MWR retrievals based on nearby radiosondes is presented in the manuscript with sole aim of demonstrating that a simple correction can be applied with just limited radiosonde data.

Since the correction method primarily reduces systematic biases (but not much random biases, Fig. 11), the results presented in Table 5 and sections 4.6 and 4.7 are similar before and after the correction, but with major differences in mean bias error (MBE) only. So, the MBE for original data is added alongside those from corrected data in Table 5.

**Major specific comments:**

1. Throughout the paper the MWR retrievals are called "measurements". Please change this.
Response: Done.

Some specific examples are reported later.

2. **Section 4.4** This section is a little bit puzzling, and I am not sure I understand it. I understand that, because the radiometer is pointing at 20-degree elevation, you may have a better chance of having less measurement degradation during rain. However, why should the radiometer do so much better during cloudy conditions than during clear sky? Or during rain than during cloudy? I assume that, if the effect of rain deposition is eliminated, the radiometer shouldn't distinguish between rainy and cloudy. To me the results could be easily explained by compensating measurement biases during precipitation, I am not sure this section should be kept at all as it is difficult to interpret.

Response: The observed larger cold biases during cloudy days than precipitation days are found to be consistent with the results by Cimini et al., 2011 whereas the larger cold biases during clear

sky days than the cloudy days are consistent with the results by Xu et al., 2015. It is not quite clear why there are such differences in the performance of the MWR during three different weather conditions, although we can speculate that better accuracies during precipitation and cloudy days than clear sky days could be due to the temperature profiles trending towards the moist adiabat and reduced temperature inversion layers. As demonstrated in the manuscript, the MWR consistently fails to detect elevated temperature inversion layers giving rise to a marked increase in cold biases in the profiles, which are more prominent on clear sky days. Due to the varying performance of the MWR under different weather conditions, there is also a possibility of shifting in the weighting function which is now under investigation.

3. **Section 4.4** How do you know what are the conditions off-zenith? The IRT is looking at zenith so how do you know whether off-zenith, where the radiometer is pointing, is cloudy or clear? Perhaps there are clouds over the radiometer and not off zenith, or the other way around. In my opinion this entire section 4.4 should be eliminated.

Response: Yes, that is correct. The three different weather condition classifications are based on the IRT looking at zenith. Since the average of two off-zenith retrievals pointing in opposite directions is used in the study, the classification based on three weather conditions are carried out with an assumption that observations from the zenith pointing IRT still hold true for off-zenith retrievals, though this assumption will lead to some limitations and uncertainties (note added in line 422 – 426). Nevertheless, the observed but varying cold biases in MWR temperature profiles during different weather conditions are found to be consistent with previous studies by Cimini et al., 2011 and Xu et al., 2015.

4. **Section 4.5** This section should be eliminated. Of course, it comes without saying, that if we do have radiosondes at the site and we correct the radiometer based on the radiosondes we get good results, we already know this. But this defeats the entire purpose of having a network of radiometers. As mentioned in my general comments, I think the manuscript would be much more valuable if the authors used the radiosondes only for evaluation and not for correction. With the correction in place the conclusion of the analysis should be that the MWR are a good "*complement*" to radiosondes and should be deployed at radiosondes sites to increase the temporal coverage but have no value at sites where there are no radiosondes.

Response: The correction of MWR retrievals based on radiosonde data is a temporary solution to minimize errors and meet our accuracy specifications discussed above. Since the linear correction for the MWR can be developed from a rather limited sample size of radiosonde data, the MWR need not be co-located near an operational NWS radiosonde site. Indeed, the fact that cold bias remains relatively steady over time, and which can be fixed with a single linear fit is surprising. We believe that this information provides other researcher with a method to replicate this simple linear correction method until a better retrieval method is available and implemented. Furthermore, our aim of presenting this correction method aligns with another reviewer who believes that it is the main novelty of this study, one can be implemented easily by other researchers.

Despite the limitations of the MWR, the MWR retrievals have been found to follow similar vertical trend to that of radiosonde profiles, particularly within the boundary layer, where degrees of freedom are much greater. The MWR profiles provide much needed real-time evolution of thermodynamic properties diurnally, that is unavailable from the twice per day radiosonde data. Therefore, the MWR has significant value at sites where there are no radiosondes (and even at sites with radiosondes) and can be a critical nowcasting tool as demonstrated by our case study and other studies such as Madhulatha et al., 2013 and Chan et al., 2011.

**Minor comments:**
1. Page 6 L138 Please change vertical resolution to vertical grid.

Response: Done (line 140)

2. Page 9 Line 209 Please change MWR measurements with MWR retrievals.

Response: Done (line 205)

3. Page 9 Line 225 and following. It is important to understand that MWR measurements have no *vertically resolved information* above roughly 2 km. It will be enough to keep the MWR comparison between 0 and 3 km as well.

Response: It is true that the accuracies of MWR retrievals decrease from the surface upward as our comparison results of the MWR retrievals with the radiosondes are better at lower heights than at higher heights. However, there are still some useful information above 2 – 3 km that are needed for deriving forecasting parameters as demonstrated in the manuscript (in our case at least up to 500 mb or 5 – 6 km). Despite limited vertically resolved information above 2 km, studies (Cimini et al., 2015, Chan et al., 2011, Madhulatha et al, 2013) have shown the value of MWR retrievals in deriving valuable thermodynamic indices that require data above 2 km.

4. Page 10 line 230: "directly measured" please change to *retrieved* as you are comparing retrievals of temperature and humidity from the MWR with those measured by radiosondes.

Response: Done (line 229)

5. Page 15 section 4.3: The high biases and RMSE of the MWR retrievals are obviously concerning and are probably due to poor calibration of the instruments combined to an inadequacy of the neural network retrievals. Both aspects can be improved operationally to lower the RMSE to less than 1.5 K between 0 and 3 km. Is the mesonet network planning to do that?

Response: Yes, the NYSM is working with the sensor vendor on improving sensor calibration and neural network retrievals. In meantime, the NYSM is using the bias correction technique as discussed in the manuscript until the application of an improved calibration and 1-DVAR technique are available. The RS based correction helps to reduce both temperature MAE and

RMSE below 2 °C within 2 km (Fig. 11). Therefore, we plan to launch limited radiosondes at the remaining 14 sites.

More importantly, our consistent results from across 3 different sites (MWR retrievals compared against two different radiosondes systems) indicate that the observed errors in the MWR retrievals are not primarily due to poor calibration or neural network performance but rather due to some inherent issues in the MWR with an ill-posed retrieval problem and rapidly fading weighting function. As mentioned before, we are exploring other retrieval algorithms, such as 1-DVAR technique.

6. Line 363-406. The discussion of the vapor density profiles would be more informative if the author could provide the range of vapor density at the sites. This would give the reader an idea of the error percentage on the profiles (for example an RMSE of 1 g/m$_3$ would be about 10% if the average vapor density at the site is 10 g/m$_3$).

Response: Across three selected sites, the mean water vapor density values are found to be 7.8 – 9.3 g m$^{-3}$ at the surface, 5.5 – 6.6 g m$^{-3}$ at 1 km and 3.9 – 4.6 g m$^{-3}$ at 2 km giving rise to error percentage of 9 – 14 % at surface, 20 – 25 % at 1 km and 26 – 33% at 2 km, with overall error of 17 – 23 % below 2 km. This information is added in line 378 – 381.

**Conclusions**

In conclusion, I think the information presented in the paper is valuable, however it would be more valuable if the authors could provide the reader with a true assessment of the value of the entire MWR network for forecast purpose. To this end the authors should not be afraid of presenting less than perfect results if that is what the MWR network is providing. Such information would be extremely valuable for those planning to deploy such networks where radiosondes are not available.

In order to evaluate the true capability of a network of radiometers for forecasting purposes the authors should use radiosondes only for evaluation and not for correction. After the uncertainties in the forecast parameters have been established (with the help of radiosondes) data from sites without radiosondes should be used to forecast events and to establish realistic uncertainties. If the results are not satisfactory the authors should discuss how the poor radiometric performance can be improved without help from radiosondes, i.e. by reviewing the calibration procedures and improving the retrievals.

We are evaluating the MWRs to determine their value for providing real-time data for weather operations and warning on high-impact weather. Specifically, we are evaluating the ability of the MWRs to provide enhanced situational awareness, a critical tool for improving short-term warnings and nowcasts. Due to the limitations of the MWR, we are in the process of understanding the issues of the MWR, which appear to be more than simply the calibration and neural network retrieval. We hope to resolve these issues with the application of better retrieval method such as 1-DVAR in near future.

---

## Author Comment (AC2)

Review of Evaluation of the New York State Mesonet Profiler Network Data

By Shrestha, Brotzge, and Wang (AMT-2022-85)

General:

This study presents an evaluation of part of the New York State Mesonet, with several stations providing microwave radiometer profiles of thermodynamic quantities and profiles of winds from a Doppler lidar. Multiple years of data are considered, and three sites located near radiosonde launch sites are evaluated by comparison to those soundings. Errors in thermodynamic quantities are large (and differ between clear-sky, cloudy, and precipitation conditions), and to resolve these errors a linear regression method is developed and applied to the MWR data. A brief case study is presented, that of a thunderstorm that would not have been characterized with the radiosonde network. Finally, some profile-related thermodynamic parameters are derived from the MWR and evaluated, but an opportunity was missed to evaluate important stability-related metrics like the Richardson number that would be available from conventional radiosonde datasets.

A manuscript like this, presenting a new, publicly available dataset, could be of interest to the AMT readership. The main novelty is the correction applied to the MWR, but insufficient detail on that correction is provided for a reader to understand the steps involved or how to implement a similar correction on a different dataset. I would recommend that the authors present this correction in more detail, and that the section on derived parameters be expanded to include the Richardson number and perhaps the boundary-layer height as well. The figures should be improved as discussed in my comments below.

Major:

1. The section on the "Correction to MWR biases" requires more detail. As lines 464-6 are written, it is not clear if the correction is derived separately for cloudy, clear, and precipitation days. Not enough information is given for a reader to attempt to replicate this correction. Are the profiles in Figure 11 taken from the 25% of the dataset used for testing, or from the 75% training? (I would presume the "testing" portion, but it is never explicitly stated.) Is this correction something that could be applied to other datasets in other locations? If so, what steps should a researcher take?

   Response: The correction is derived separately for precipitation, cloudy and clear sky days for both temperature and water vapor density. Profiles in Fig. 11 are only from the testing dataset (25% of total dataset for each weather conditions). The paragraph (line 475 – 486) is revised for better clarity.

   Our results have shown that limited clear sky days radiosonde data are found to be helpful to reduce biases in MWR retrievals. Therefore, the NYSM is planning to launch several radiosondes to bias correct remaining 14 MWRs in summer of 2022. As a long-term plan, one dimensional variational (1-DVAR) technique (Cimini et al., 2011) is under consideration and will be implemented in near future.

2. Some profile-related thermodynamic parameters are derived from the MWR profiles and briefly evaluated in section 4.6. Because both winds and thermodynamics are available from the RS, why not combine the MWR and the DL datasets to calculate stability metrics like the bulk Richardson number (Ri)? That would be a very useful test of the utility of the network. An important opportunity was missed here. The Ri could have been incorporated into the discussion of the thunderstorm case study as well. Similarly, the planetary boundary layer (PBL) height is easily calculated from the DL dataset, and that could have been compared to sounding-based estimates as well. Or, if the radiosonde dataset is not adequate for calculating the PBL height, pointing out the utility of the new mesonet capabilities could be a nice addition to the paper.

Response: The single value bulk Richardson number used in convective storm forecasting that depends on CAPE and deep layer wind shear, 0 – 6 km AGL, (Weisman and Klemp, 1986; Evenson, 1993, https://glossary.ametsoc.org/wiki/Bulk_richardson_number) is not possible to derive due to lack of DL wind measurements at 6 km.

It is possible to derive vertical profile of Ri (Sorenson et al. 1998) and PBL height; however, as the RS launch times (7 am and 7 pm LT) are not optimal times for the DL data availability, those two parameters are not included in this study. However, the authors plan to derive and compare those two parameters when we launch our own radiosondes (during the afternoon) to bias correct the remaining 14 MWRs. The PBL height using Ri and other DL methodologies will be extensively discussed in our next paper. The utility of the NYSM Profiler Network to derive those parameters is briefly mentioned in this manuscript (line 530-531)

Minor:

1. Line 133: can you comment on how often the lidar can actually retrieve wind estimates from 7000m? We see later that almost no data is available above 3km, so that should be noted here.

   Response: The DL data availability is very limited and rarely available above 3 km. There are occasional data availability above 3 km when long range transported wildfire smoke is detected in the region during the summer months. This information is added in line 133-135.

2. Figure 2 would be more intuitive with height on the y-axis. The colors are difficult to distinguish (especially for red-green colorblind readers) so please consider using different line styles.

   Response: Done. Fig. 2 is revised with different colors and height on the y-axis.

3. Lines 200-205: how sensitive is the agreement between the radiosondes and the doppler lidar to the averaging time selected?

Response: Using ±10 min (20 minutes averaging) and ±30 min (60 minutes averaging) centered at the radiosonde launch time produced slightly better DL/MWR comparison results ($R^2$ increased by 1-2%) than using only +10 minutes and +30 minutes starting from the launch time.

4. Near line 210: please explicitly state which RS sites were used to train the neural nets for each of the MWR sites

Response: The RS sites used to train the neural network for three selected MWR sites are listed in line 210-212. The list of NWS RS sites used to train neural networks for all 17 MWR sites are as follows:

| NYSM Site | NWS RS #1 | NWS RS #2 |
|---|---|---|
| Albany | Albany | |
| Belleville | Albany | Buffalo |
| Bronx | Upton | |
| Buffalo | Buffalo | |
| Chazy | Albany | |
| Clymer | Buffalo | Pittsburgh |
| East Hampton | Upton | |
| Jordan | Albany | Buffalo |
| Owego | Albany | Munich (Germany) |
| Queens | Upton | |
| Red Hook | Albany | |
| Staten Island | Upton | |
| Stony Brook | Upton | |
| Suffern | Albany | |
| Tupper Lake | Albany | Munich (Germany) |
| Wantagh | Upton | |

5. Line 252: better to provide a textbook reference than a link that disappear over time.

Response: Paper references added in addition to web link (line 249-251).

6. Fig 3, 6: again, please don't rely on red-green differences. Use colors that are more easily distinguishable or also use line style differences

Response: All those figures are revised accordingly.

7. Why doesn't Fig 4 also include Stony Brook?

Response: DL data from Stony Brook was not available for same date and time as for other two sites. Stony Brook DL data from different date is added in Fig. 4.

8. Line 347: explicitly point out the elevated inversion layer near 1 km in Fig7a, near 2.5km in Fig7b, etc.

   Response: The heights corresponding to the elevated inversion layers are added in line 347-348)

9. Line 459-460: write these statements out explicitly instead of using the confusing parenthetical formulation.

   Response: Done (line 469-470)

10. Figure 12 relies on the rainbow color table although extensive literature is available showing that it is suboptimal (Light and Bartlein, 2004; Stoelzle and Stein, 2021)

    Response: Thank you very much for the suggestion! We have made extra efforts to test different color schemes and replaced rainbow color map with plasma as suggested by Stoelzle and Stein, 2021.

References

Light, A. and Bartlein, P. J.: The end of the rainbow? Color schemes for improved data graphics, Eos Trans. AGU, 85, 385–391, https://doi.org/10.1029/2004EO400002, 2004.

Stoelzle, M. and Stein, L.: Rainbow color map distorts and misleads research in hydrology – guidance for better visualizations and science communication, Hydrology and Earth Systems Sciences, 25, 4549–4565, https://doi.org/10.5194/hess-25-4549-2021, 2021.

**Citation**: https://doi.org/10.5194/amt-2022-85-RC2

---

## Referee Report (RR1)

Review for: ***"Evaluation of the New York State Mesonet Profiler Network Data"*** (amt-2022-85) submitted to *Atmospheric Measurement Techniques* by *Bhupal Shrestha et al.*

**General recommendation:**

Major revision

**Synopsis:**

The manuscript presents the observations by a network of microwave radiometer, Doppler lidar and all-sky camera in New York state. For three of the 17 stations, comparisons with nearby radiosondes have been performed and are analysed in terms of profiles of temperature, humidity and wind as well as for different thermodynamic indices that were derived from the original profiles. Furthermore, the aerosol optical depth from the camera observations is evaluated using AERONET sun photometer observations.

**General comments:**

The presentation of this quite dense network of ground-based remote sensing observations will be very useful to many readers if some more information is given on the quality of the dataset, especially regarding the microwave radiometer (MWR) dataset.

The Doppler lidar observations seem to agree well with the radiosondes. Sections 4.1 and 4.2 are well presented and can be published like this.

However, I am concerned for the large biases in the MWR observations compared to the radiosondes (especially for temperature). This cannot be only explained by retrieval uncertainties, especially the 8 K bias for clear sky cases. The reason for these large biases must be further investigated before the manuscript is ready for publication.

It is well known from literature that the quality of MWR temperature profiles decreases significantly with height. Therefore, it might be useful to limit the evaluation of the MWR profiles to the lower troposphere (e.g. 3 km), as for wind profiles.

Some more detailed questions and comments to the retrievals and their application:

- How were the neural network retrievals trained that you use for the MWR profiles? You are presenting the retrievals in lines 205ff., but I am wondering whether a retrieval self-test (by applying to forward-modelled radiosonde profiles) would result in similar biases. Do you have any information about that? If the self-test produces also the observed biases then there is an inherent problem with the retrieval (e.g. the setup of the radiative transfer model, the gas absorption models used, or the neural network design). On the other hand, if the self-test is bias-free then the reasons for the bias must stem from the instrument (e.g. wrong channel bandpasses, faulty calibration, etc.). However, the second hypothesis is much less likely, because in this case, the performance would not show such similar bias/error patterns at the different locations.

- Do you apply separate retrievals for 90° and 20° elevation observations? Or do you have any combined retrieval which uses brightness temperatures at both angles as input? How do the zenith and off-zenith observations compare e.g. during clear sky? I suppose that there is a stronger bias in the off-zenith observations as the optical path through the atmosphere is

much longer and there is less contribution to the signal from the higher levels due to attenuation (esp. for temperature profiles).

- The problem due to drifting radiosondes is rather minor and should not affect the retrieval bias (only the RMSE) as the sondes might randomly drift to warmer/colder/drier/moister regions. (see also p.19, l. 405-406).

- Concerning the calibration of the instruments: How often were the radiometers calibrated using liquid nitrogen? Do you see jumps in the performance before and after calibrations?

- I am a bit concerned by the bias correction. I am sure that on average you are improving the profiles, but how do you know that e.g. in severe weather situations (which are often at the edge of the statistical distribution), the bias correction performs well?

- For the bias correction, you are separating your dataset in clear sky/cloudy/rainy cases (Fig. 10a,b,c). What is the variability of these biases as a function of height for each of the three weather classes? Are there any seasonal differences? What about very cold winter days or hot summer conditions? I am pretty sure that the bias is quite variable, and the mean bias of 0°C / 0 gm$^{-3}$ in Fig. 11 (a) and (d) does not show the range or the uncertainty of possible biases. Also, I would like to know more about the physical reason of the different biases depending on the weather.

- Usually, a bias correction is done before applying the retrieval, i.e. a correction of the brightness temperatures (as it is done e.g. in Lohnert and Maier, 2012), but I guess that this is beyond the scope of this paper.

**Detailed comments:**

- p.1, l.20-21: Why do you only mention biases here?
- p.3, l.81: The RMSE of <= 7 K is a very high number. Other publications show quite low numbers.
- p.6, l.140-141: The vertical resolution is determined by the retrieval algorithm. The "true" vertical resolution is very coarse and decreasing with height (see e.g. Crewell and Lohnert, 2007). For these passive observations, often the degrees of freedom are determined, which is between 2 and 5 for the whole profile (depending on the choice of frequencies and angles).
- p.6, table 1: Are you sure that the manufacturer gives a relative humidity accuracy of 2% ?
- p.16, fig. 6 a-c: These scatterplots are not very useful, as the performance of the MWR retrievals depends strongly on the height above ground (and not on the temperature itself). You could try to make a color plot Temperature bias vs. height for bins of 1°C vs. 100m (or the vertical resolution of the retrieval)
- p.19, l.396-397: Yes, I can see that the biases are significant, but you should further investigate the reason for that.
- p. 19, l. 398ff.: *"This is likely because the MWR measured V-band observations are ingested into the neural network with greater weighting function at lower heights than at higher heights to produce a finer vertical resolution at lower heights."*
  This phrase makes no sense to me. The neural network itself will determine the weights of the single V-band brightness temperature observations for the temperature profile. This has nothing to do with the vertical resolution.
- p.19, l.401-402: *"In contrast, the water vapor density comparisons show better results at higher altitudes than at lower altitude"*

This is not surprising, as there is not much water vapor (absolute) in upper levels. If you compared the relative error, the result would look very different.

- p.20, table 4: This table would be more interesting as a function of height (or limited to the lower troposphere, such as up to 3 km)

---

## Author Response (AR2)

Review for: ***"Evaluation of the New York State Mesonet Profiler Network Data"*** (amt-2022-85) submitted to *Atmospheric Measurement Techniques* by *Bhupal Shrestha et al.*

**General recommendation:**
Major revision

**Synopsis:**
The manuscript presents the observations by a network of microwave radiometer, Doppler lidar and all-sky camera in New York state. For three of the 17 stations, comparisons with nearby radiosondes have been performed and are analysed in terms of profiles of temperature, humidity and wind as well as for different thermodynamic indices that were derived from the original profiles. Furthermore, the aerosol optical depth from the camera observations is evaluated using AERONET sun photometer observations.

**General comments:**
The presentation of this quite dense network of ground-based remote sensing observations will be very useful to many readers if some more information is given on the quality of the dataset, especially regarding the microwave radiometer (MWR) dataset.

The Doppler lidar observations seem to agree well with the radiosondes. Sections 4.1 and 4.2 are well presented and can be published like this.

However, I am concerned for the large biases in the MWR observations compared to the radiosondes (especially for temperature). This cannot be only explained by retrieval uncertainties, especially the 8 K bias for clear sky cases. The reason for these large biases must be further investigated before the manuscript is ready for publication.

Response: The overall extent of observed cold biases in the MWR temperature are consistent with previous studies by Cimini et al., 2011, Xu et al., 2015 and Cimini et al., 2015. These studies have attributed such cold biases to varieties of reasons such as neural network performance, calibration uncertainty, gas absorption model biases, radiosonde biases and radiosonde drift distances, though latter two might have less significant effect. The larger cold biases during cloudy days than precipitation days are consistent with the results by Cimini et al. (2011) whereas the larger cold biases during clear sky days than the cloudy days are consistent with the results by Xu et al. (2015).

Due to the near identical error profiles from three selected NYSM Profiler sites, those observed biases could be sensor specific, rather than poor neural network performance. The issue of cold biases has not been fully understood yet, and it is still under investigation by our team in consultation with the manufacturer, Radiometrics. However, we speculate that significant biases inherent to the brightness temperature could be a major factor for large temperature biases (For further details, please refer to responses to the first detailed question) and the resulting height dependent biases could be the result of rapidly fading weighting function. The Radiometrics has developed automatic calibration (Acal) technique that replaces liquid nitrogen calibration and its associated uncertainty. Our preliminary analysis of the application of Acal at Albany has shown quite an improvement in the retrievals and is therefore, under extensive investigation at multiple NYSM sites. We are also evaluating one-dimensional variation (1-DVAR) technique (Cimini eta al., 2011) to improve the MWR retrievals. While those techniques are under evaluation and consideration for long-term use, the radiosonde correction method discussed in the manuscript is shown as a temporary solution to minimize such observed biases.

It is well known from literature that the quality of MWR temperature profiles decreases significantly with height. Therefore, it might be useful to limit the evaluation of the MWR profiles to the lower troposphere (e.g. 3 km), as for wind profiles.

Response: We agree that it might be useful to limit the evaluation of the MWR profiles within 3 km similar to that of DL profiles. However, based on our data requests throughout the profile, we think that our data users would like to see MWR retrievals and their associated errors along the profile and hence, presented our analysis to the maximum range of 10 km. Please find the further comments at the end.

Some more detailed questions and comments to the retrievals and their application:

• How were the neural network retrievals trained that you use for the MWR profiles? You are presenting the retrievals in lines 205ff., but I am wondering whether a retrieval self-test (by applying to forward-modelled radiosonde profiles) would result in similar biases. Do you have any information about that? If the self-test produces also the observed biases then there is an inherent problem with the retrieval (e.g. the setup of the radiative transfer model, the gas absorption models used, or the neural network design). On the other hand, if the self-test is bias-free then the reasons for the bias must stem from the instrument (e.g. wrong channel bandpasses, faulty calibration, etc.). However, the second hypothesis is much less likely, because in this case, the performance would not show such similar bias/error patterns at the different locations.

Response: The neural networks are trained using the historical radiosonde data from a site that is close to MWR site in terms of climatology and elevations. The three selected NYSM Profiler site MWR are very close to NWS radiosonde site and hence, those MWRs at Buffalo, Albany and Stony Brook are trained using NWS radiosonde data at Buffalo, Albany, and Upton.

We did the self-test for one month of the data at Albany and found out significant cold biases in V-band brightness temperature that is consistent with Ware et al., 2013, which could be due to instrumental and/or calibration issues. It is also reported that absorption models have some biases in each V-band channel (Hewison 2007). Similarly, significant V-band brightness temperature biases has also been reported immediately following liquid nitrogen calibration (Lohnert et al., 2012, Illingworth et al., 2019). Therefore, we agree that there is an inherent problem with the retrieval, and we are actively investigating it. These potential issues are mentioned in the paragraph of line 401-422.

As mentioned above, our preliminary analysis of Acal technique developed by Radiometrics has shown improvements in the MWR temperature retrievals. Therefore, we are extensively evaluating Acal technique along with 1-DVAR technique to reduce the MWR retrievals biases for the operational use, mentioned in the summary and conclusions line 689-693.

• Do you apply separate retrievals for 90° and 20° elevation observations? Or do you have any combined retrieval which uses brightness temperatures at both angles as input? How do the zenith and off-zenith observations compare e.g. during clear sky? I suppose that there is a stronger bias in the off-zenith observations as the optical path through the atmosphere is much longer and there is less contribution to the signal from the higher levels due to attenuation (esp. for temperature profiles).

Response: There are two separate retrievals for zenith (90°) and two off-zenith (20° facing North and South) observations. Since extensive study was carried out by Xu et al., 2014 showing the off-zenith retrievals better than zenith retrievals not only during precipitation but also during non-precipitation days, we followed their analysis and only used average of two off-zenith retrievals in this study. Based on our limited study between off-zenith and zenith retrievals, we agree with Xu et al., 2014 that off-zenith retrievals are better in precipitation days while those

retrievals are somewhat comparable to zenith retrievals during non-precipitation days, though the off-zenith retrievals have reduced lower V-band weighting functions as compared to zenith retrievals (the zenith weighting function of 2km is reduced to ~0.8 km for off-zenith).

• The problem due to drifting radiosondes is rather minor and should not affect the retrieval bias (only the RMSE) as the sondes might randomly drift to warmer/colder/drier/moister regions. (see also p.19, l. 405-406).

Response: We totally agree that problem due to drifting radiosondes is minor and significant contribution in the retrieval biases could be due to error inherent to the MWR, discussed above. The sentence is reworded accordingly in line 417.

• Concerning the calibration of the instruments: How often were the radiometers calibrated using liquid nitrogen? Do you see jumps in the performance before and after calibrations?

Response: As recommended by the manufacturer Radiometrics, liquid nitrogen calibration is done once every 6 months. We do see some differences in the performances before and after the calibrations within 2-5%. More importantly, due to some biases observed immediately after liquid nitrogen calibration as also reported by Lohnert et al., 2012, the Radiometrics developed automatic calibration (Acal) that will eventually replace liquid nitrogen calibration is under evaluation by the NYSM team for operational use. As mentioned earlier, our preliminary results showed improvement in retrievals using Acal.

• I am a bit concerned by the bias correction. I am sure that on average you are improving the profiles, but how do you know that e.g. in severe weather situations (which are often at the edge of the statistical distribution), the bias correction performs well?

Response: The radiosonde-based correction method significantly improves the profiles (precisely temperature) for all three different types of weather conditions. The differences in the performances of corrected and uncorrected MWR retrievals and corresponding indices in the event of severe thunderstorm are demonstrated in the case study in Section 4.7 and Fig. 13, and the corrected data clearly outperforms the uncorrected data. Overall, the corrected data helps in meeting the threshold criteria set by the NWS. Further evaluation and statistical analysis of MWR retrievals (both corrected and uncorrected) in nowcasting severe weather is a part of ongoing study.

It is also to be noted that radiosonde correction method discussed in the manuscript is a temporary solution and will be replaced in near future by Acal and/or one dimensional variational (1-DVAR) technique for long-term use that has shown significant improvement in MWR data (Cimini et al., 2011). Since our studies have shown limited clear sky radiosonde data can help to reduce biases in MWR retrievals, we believe that this information provides other researchers/users with a method to replicate this simple linear correction method until a better method is available and implemented.

• For the bias correction, you are separating your dataset in clear sky/cloudy/rainy cases (Fig. 10a,b,c). What is the variability of these biases as a function of height for each of the three weather classes? Are there any seasonal differences? What about very cold winter days or hot summer conditions? I am pretty sure that the bias is quite variable, and the mean bias of 0°C / 0 gm-3 in Fig. 11 (a) and (d) does not show the range or the uncertainty of possible biases. Also, I would like to know more about the physical reason of the different biases depending on the weather.

Response: We have updated our error profile plots with error bars representing one standard deviation (Fig. 3, 6, 8 and 10) to show the variability of the biases as a function of height. We have observed seasonal differences in MWR data that are relatively larger during winter than summer, but those seasonal differences were not as pronounced as those seen during different weather conditions. Therefore, we decided to present the classification of the MWR data based on three weather conditions only. Since the radiosonde correction method is a temporary solution (and not to make plots crowded), we did not include error bars in Fig. 11 as compared to other previous plots. We believe it is quite evident with differences in MBE, MAE and RMSE before and after correction that radiosonde-based correction method can be a useful application.

We are not quite clear on why there are different biases depending on three different weather conditions. However, we speculate that better accuracies of temperature during precipitation and cloudy days than clear sky days could be due to the temperature profiles trending towards moist adiabat and reduced temperature inversion layers. As shown in Fig. 7, the MWR temperature profiles fail to detect elevated temperature inversion layers giving rise to a marked increase in cold biases that are more prominent on clear sky days. This physical reasoning is added in line 458-460. The better accuracies of water vapor density during clear sky days could be due to lower variability of the moisture than cloudy/precipitation days.

• Usually, a bias correction is done before applying the retrieval, i.e. a correction of the brightness temperatures (as it is done e.g. in Lohnert and Maier, 2012), but I guess that this is beyond the scope of this paper.

Response: Yes, this is beyond the scope of this paper.

**Detailed comments:**

• p.1, l.20-21: Why do you only mention biases here?

Response: Not any specific reason. Just do not want to overwhelm reader with too many error numbers in the abstract.

• p.3, l.81: The RMSE of <= 7 K is a very high number. Other publications show quite low numbers.

Response: The RSME up to 7 K is the maximum value observed along the profile reported in those publications cited, though few have reported much lower RMSE than 7 K. Since the performance of the MWR depends upon weather conditions as well as seasons, those publications show marked variations in errors, which could be due to different selected case study period.

• p.6, l.140-141: The vertical resolution is determined by the retrieval algorithm. The "true" vertical resolution is very coarse and decreasing with height (see e.g. Crewell and Lohnert, 2007). For these passive observations, often the degrees of freedom are determined, which is between 2 and 5 for the whole profile (depending on the choice of frequencies and angles).

Response: We agree that the vertical resolution is determined by the retrieval algorithm, which depends on inverse method and atmospheric conditions and degrades with height, as also reported by Hewison, 2007 and Cimini et al., 2015. To be more precise, we have changed "measurement" to "the retrieved profile" in line 142.

• p.6, table 1: Are you sure that the manufacturer gives a relative humidity accuracy of 2%?

Response: Based on the Radiometrics manual, temperature accuracy is up to 2°C and relative humidity accuracy is up to 2% at the surface, that tend to decrease along the height. This note is added in the table for clarity.

• p.16, fig. 6 a-c: These scatterplots are not very useful, as the performance of the MWR retrievals depends strongly on the height above ground (and not on the temperature itself). You could try to make a color plot Temperature bias vs. height for bins of 1°C vs. 100m (or the vertical resolution of the retrieval)

Response: We agree that the performances of the MWR retrievals are height dependent. However, the sole purpose of showing these scatterplots is to demonstrate an overall one-to-one relationship for providing the instant shape of the relationship between MWR and RS data cumulatively from all possible height levels. For any information on the errors along the height, mean bias error (MBE) with one standard deviation error bars, mean absolute error (MAE) and root mean squared error (RMSE) are presented alongside for specific height dependent error information.

• p.19, l.396-397: Yes, I can see that the biases are significant, but you should further investigate the reason for that.

Response: This is already discussed above. The changes are made accordingly in the paragraph (line 401-422). As already mentioned above, the exact reason for the significant biases is under investigation, but we believe it is most likely due to instrumental and calibration issues that inherently introduce biases in the brightness temperature.

• p. 19, l. 398ff.: *"This is likely because the MWR measured V-band observations are ingested into the neural network with greater weighting function at lower heights than at higher heights to produce a finer vertical resolution at lower heights."* This phrase makes no sense to me. The neural network itself will determine the weights of the single V-band brightness temperature observations for the temperature profile. This has nothing to do with the vertical resolution.

Response: The better accuracy of MWR retrieved temperature profiles in the lower altitude (within the boundary layer) than higher altitude is due to the weighting function of V-band channels, peaking near the surface and fading rapidly above the boundary layer (Hewison, 2007, Cimini et al., 2011, Westwater, 1993). Similarly, it could also be due to the fact that the temperature information is concentrated in the lowest few kilometers but drops off steadily with height (Hewison 2007). So, we agree that the sentence is not precisely correct and has been changed with the above information in line 405-407.

• p.19, l.401-402: *"In contrast, the water vapor density comparisons show better results at higher altitudes than at lower altitude"*

This is not surprising, as there is not much water vapor (absolute) in upper levels. If you compared the relative error, the result would look very different.

Response: We totally agree on that. Sentence is reworded in line 409-410.

• p.20, table 4: This table would be more interesting as a function of height (or limited to the lower troposphere, such as up to 3 km)

Response: Without a doubt, it is true that the accuracies of MWR retrievals decrease from the surface upward. It may be enough to keep the comparison results within 3 km. But, because we are also deriving forecast indices that require data from at least up to 500 mb or $5 - 6$ km height, we decided to present data from all the heights. For any specific height dependent errors, vertical profiles of MBE, its statistical significance and one standard deviation error bars along with MAE and RMSE are presented, which provides readers with the height dependent errors, though we agree that overall comparison results would improve by considering height up to 3 km only. Moreover, being a leading profiler network in the nation, we thought that we should present analysis throughout the profiles for the readers and for those planning to deploy such networks in near future so that they are aware of advantages/disadvantages of deploying DL/MWR and the quality of data along the height from diverse geographical locations. Furthermore, the data requests from our users have shown interest in data along the profile and have frequently asked about the accuracy of data throughout the profile.

---

## Author Response (AR3)

Submitted on 07 Sep 2022
Anonymous referee #3

General comments:
There are a few points that need to be taken into account before the manuscript can be published.

The authors would like to thank the reviewer for the time and effort to review this manuscript and is very much appreciated. Please find the responses to each comment below.

Detailed comments:
-In plots 3,6,8,10, the values of MBE (and others) in the layers close to ground can't be clearly distinguished. You should think of another way to present the data, maybe by removing some height layers? In addition, the differentiation between p>0.05 and p<0.05 is not very useful here. For sake of simplicity, you could just use the same color and don't differentiate whether the bias is significant or not.

Response: For clarity of error profiles at the lower heights, Fig. 3 is replotted up to 1 km and Figs. 6, 8 and 10 are replotted up to 2 km only and are submitted in the Supplement file (Fig. S1 – S4).

- Lines 206-208: As you are using only off-zenith observations, the larger biases for temperature profiles compared to other studies may also arise from the fact that you didn't use zenith observations. See e.g. Cimini et al. 2011 or Cimini et al., 2015, who show that zenith retrievals might provide smaller biases. This should be mentioned in the discussion.

Response: For limited period, we compared the MWR retrievals between off-zenith and zenith retrievals and we found out that off-zenith retrievals are clearly better during precipitation days, consistent with Xu et al., 2014, while those retrievals are somewhat comparable to zenith retrievals during non-precipitation days. As a result, we decided to use off-zenith retrievals in this study. But we agree that there could be comparatively lower biases for zenith retrievals during non-precipitation days. This note is added to the line 207-208.

- Lines 415-422: This part needs to be rephrased, I think there is still some old text in the new version.

Response: The suggested part is rephrased in line 418-422.

- Line 691: You mentioned the newly Acal method by Radiometrics which will improve the calibration. Can you just give some information on the principle that is used for this calibration method, if it doesn't involve liquid nitrogen? Is it still an absolute calibration that is independent of external data sources? This should be shortly mentioned in the discussion.

Response: Automatic calibration (Acal) continuously evaluate and calibrate the instrument and reduces the biases in the brightness temperature compared to biannual liquid nitrogen calibration. Acal does not require access to internet and/or other data sets otherwise not embedded in the software. We do not have further information from the Radiometrics about Acal as of now as they are planning to release details very soon in about a couple of months. To answer your questions, as a courtesy Radiometrics provided us the information about Acal to share as below:

> *"ACal utilizes an ensemble of independent, exploitable, physically constrained atmospheric relations that - when imposed simultaneously - provide a verifiably accurate framework by which the near-entirety (> 3 sigma) of the site-specific brightness temperature joint probability distributions can be fully characterized such that measured quantities can be statistically compared to expected quantities for the purposes of establishing and correcting calibration drift".*

As mentioned in the manuscript, we have been testing Acal results extensively at multiple sites, and we are excited about the results that significantly reduces cold biases in the MWR retrievals.

---

## Author Response (AR4)

**Comments to the author**:
Dear Authors,

I am happy to report that your manuscript is being accepted for publication in AMT.

I nevertheless ask you to include a sentence, maybe in Section 4.5 (Correction to MWR biases) and Section 5 (Summary and conclusions), where you mention the alternative possibility to apply a bias correction to the MWR brightness temperatures before applying the retrieval methodology (as it is done e.g. in Lohnert and Maier, 2012), which you can say goes beyond the scope of this paper.

Thanks and Best Regards,
Laura Bianco
Associate Editor AMT

Response: Thank you for accepting this manuscript! Really appreciate your help!

And again, thank you for your suggestion. The sentence is added both in Section 4.5 (line 519-522) and Section 5 (line 693-695).